# Inter-District Road Infrastructure and Spatial Inequality in Rural Indonesia

Ribut Nurul Tri Wahyuni [1,2,*], Mohamad Ikhsan [1], Arie Damayanti [1] and Khoirunurrofik Khoirunurrofik [1]

1   Faculty of Economics and Business, Universitas Indonesia, Depok 16424, Indonesia
2   STIS Polytechnic of Statistics, Jakarta 13330, Indonesia
*   Correspondence: rnurult@stis.ac.id

**Abstract:** Road quality plays an important role, especially in rural areas where most poor households are situated. This study aims to calculate the Rural Access Index (RAI), an indicator of rural road quality (SDG indicator 9.1.1), at the district level, to evaluate the implementation of the Nawacita programme in Indonesia from 2014–2020. The RAI describes the proportion of rural residents who live within 2 km of an all-season road. This study recommends the utilisation of road network maps, urban–rural boundary maps, three road network condition datasets, and WorldPop data to calculate the RAI. The results show that during this period, the RAI increased and its inequality decreased, specifically in the regions of priority for this programme (Papua and West Papua). The results also capture a strong pattern of regional convergence. To ensure the future success of this implementation, the government can create regulations to designate several road infrastructure projects as a national strategy, as well as increase tax collection and private sector investment as sources of road infrastructure development funding.

**Keywords:** Rural Access Index; all-season road; inequality; regional convergence





## 1. Introduction

Limited road connectivity can result in high transportation costs and long travel times, which may impact sectoral productivity (Bell and van Dillen 2014; Haughton and Khandker 2009), employment (Mu and Van de Walle 2011) and poverty (Dercon et al. 2012; Khandker and Koolwal 2011). A lack of access to the outside market, for instance, makes it difficult for people to find new jobs and discourages investment, especially in rural areas where most poor households are situated.

Roberts et al. (2006) estimated that 68.3 per cent of rural residents lack access to the global road network. Almost a billion people reside in rural areas without access to paved national roads (Asher and Novosad 2020). As shown in Table 1, in 2011, 43.27 per cent of Indonesia's rural areas did not have access to paved road networks. Rural road construction was also unequal. In eastern Indonesia, 77 per cent of rural areas lacked access to paved roads connecting villages. Similarly, 62.16 per cent of Borneo Island's rural areas lacked access to paved roads connecting villages. Other islands had paved roads connecting villages in less than 46 per cent of rural areas.

The government has been implementing the Nawacita programme by reducing fuel subsidies since 2014 to boost infrastructure development (Salim and Negara 2018). This policy prioritises accelerating connectivity between peripheries and growth centres so that inter-regional inequality can be reduced, particularly in rural areas and eastern Indonesia (Bappenas 2014). State spending on infrastructure has increased significantly, from 8 per cent of the total state budget in 2014 to 19 per cent of the total state budget in 2017. Moreover, the President of Indonesia has created the Committee for the Acceleration of Priority Infrastructure Delivery (KPPIP), a special task force with the responsibility of coordinating policies among various stakeholders and unblocking stalled national strategic projects and priority projects (Salim and Negara 2018). In the 2015–2019 National

Medium-Term Development Plan, the government committed to building 2600 km of roads. To balance the geographic concentration of investment, at least half of the government expenditure went to areas outside the capital region (Bappenas 2014), such as outside Java.

**Table 1.** The percentage of Indonesian rural areas by inter-village road condition and regional group.

| Regional Group | Zone | 2011 | | 2014 | | 2020 | |
|---|---|---|---|---|---|---|---|
| | | Paved | All-Season | Paved | All-Season | Paved | All-Season |
| Sumatra | Western | 54.16 | 88.30 | 58.86 | 84.11 | 78.05 | 91.23 |
| Java | Western | 78.80 | 97.69 | 84.04 | 97.00 | 95.21 | 98.63 |
| Bali and Nusa Tenggara | Central | 57.16 | 88.40 | 61.71 | 85.98 | 75.53 | 92.87 |
| Borneo | Central | 37.84 | 68.92 | 42.27 | 66.96 | 57.40 | 72.10 |
| Sulawesi | Central | 60.04 | 87.55 | 65.66 | 88.47 | 82.09 | 91.93 |
| Moluccas and North Moluccas | Eastern | 39.49 | 55.70 | 54.21 | 65.80 | 61.96 | 68.74 |
| Papua and West Papua | Eastern | 16.84 | 32.51 | 26.39 | 39.40 | 29.59 | 39.32 |
| Indonesia | | 56.73 | 83.34 | 63.56 | 83.87 | 77.76 | 87.78 |

Source: Author's calculation from The Potensi Desa (Podes) survey data, BPS-Statistics Indonesia.

After the Nawacita programme's implementation, access to paved inter-village roads in rural areas grew significantly. The percentage of Indonesia's rural areas that did not have access to paved road networks fell to 43.27, but road inequity persisted. Eastern Indonesia has lagged behind western Indonesia in terms of rural road infrastructure development. Unfortunately, information about Indonesian rural roads' connectivity and inequality to support this opinion, other than the data in Table 1, is currently unavailable.

Few regional indicators measure rural road connectivity correctly. Conventional measurements are total road length and the proportion of paved roads (Iimi et al. 2016), which are not good predictors for rural roads (World Bank 2016). These indicators barely change over time, although the government has spent a lot of money upgrading the road network (Iimi et al. 2016). The quality of roads is often unknown and a matter of concern in developing countries (World Bank 2016). In Indonesia, besides total road length and the proportion of paved roads, the government uses steady-road condition data to indicate road connectivity. These data are only available for the national road network by province, without rural–urban separation. They are calculated from the International Roughness Index (IRI) and used as an indicator of sustainable development goals (SDGs), namely 9.1.1 (Bappenas 2017, 2020), even though the United Nations (UN) recommendation uses the Rural Access Index (RAI).

The objectives of this study were to calculate the RAI and its regional inequality. The RAI was used as an indicator of rural road connectivity in Indonesia. It shows the proportion of rural residents who live within 2 km, usually equal to a walk of 20–25 min, of an all-season road. The term "all-season road" refers to a road that is drivable all year round by the prevailing rural transport mode (Iimi et al. 2016; Roberts et al. 2006; Workman et al. 2019; World Bank 2016). The RAI is a new rural road connectivity measurement method based on Geographic Information Systems (GIS) data. This method resolves the limitations of conventional measurements. Iimi et al. (2016) and Mikou et al. (2019) calculated the RAI by utilising rural population distribution data from WorldPop or LandScan and road network data from the government or OpenStreetMap (OSM).

The best policies for rural road access improvement require estimates for local regions, such as at the district level. This is the first study conducted in Indonesia to provide such estimates. Because the Nawacita policy places a high priority on reducing inequality in certain areas (e.g., eastern Indonesia), this study also provides rural road connectivity inequality by regional group. Indonesia is divided into seven regional groups, each with multiple provinces. Each province has a number of districts, and each district consists of several subdistricts, which include rural and urban areas. National roads are under the authority of the central government and connect the capitals of the provinces. The provin-

cial government has the jurisdiction to construct provincial roads connecting provincial capitals to district capitals. Finally, the district government is responsible for managing local roads. Because of data limitations, this study used only national and provincial roads to calculate the RAI.

This study aims to identify districts with poor rural road quality and regional groups with high rural road inequality. With these data, the government can evaluate the effects of the Nawacita programme and determine priority regions for rural road construction.

## 2. Methodology

The first step was to calculate the RAI at the district level. The RAI needs several datasets: population distribution maps, urban–rural classification data, village maps, road maps, and road network condition data. Step-by-step procedures for calculating the RAI are shown in Figure 1.

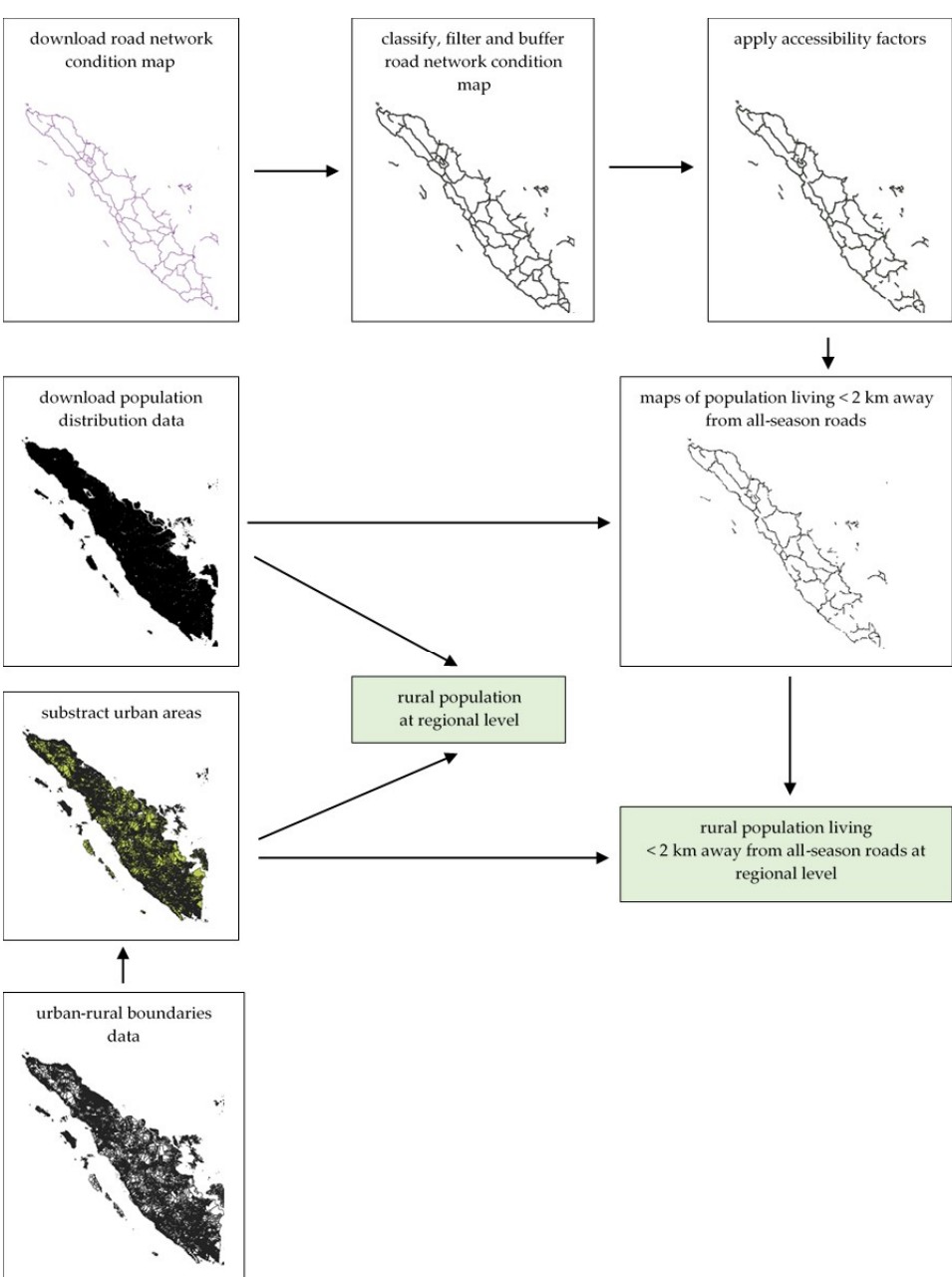

**Figure 1.** Overview of step-by-step procedures for calculating the RAI.

We used population distribution maps from WorldPop, which is the most robust dataset available, according to Mikou et al. (2019) and World Bank (2016). WorldPop uses the latest national census data and other data from countries to produce 100 m × 100 m of population distribution data. It can be downloaded freely and used in QGIS software (Workman et al. 2019). This study also conducted a robustness check by using 1 km × 1 km of population distribution data from LandScan.

We applied the urban–rural classification data from the Regulation of the Head of BPS-Statistics Indonesia Number 37 Year 2010. Then, we combined the village map with the urban–rural classification data. From this combination, we chose only rural areas to create the rural map. The intersection of the population distribution data and the rural map results in the total rural population.

This study utilised a road map from the Directorate General of Highways. Although Iimi et al. (2016), Li et al. (2022), Mikou et al. (2019) and Workman et al. (2019) recommend using OSM data, this study did not utilise it because Indonesian OSM data from 2014 to 2020 were inconsistent. This inconsistency is shown in Figure A1. We then buffered the road map with a 2-km radius.

All-season road identification uses data from the Directorate General of Highways and refers to paved roads with an IRI of less than 6 m per kilometre, unpaved roads with an IRI of less than 13 m per kilometre, paved roads in excellent, good, or fair condition, and unpaved roads in excellent or good condition (Workman et al. 2019). We also used Podes survey data from BPS-Statistics Indonesia for all-season road identification, specifically the existence of inter-village roads that can be traversed by motorised vehicles with four or more wheels throughout the year. This study applied all methods to specify all-season national roads in 2018. The results show that when data on the surface type and roughness of regional roads are not available, we can use the last method as a substitute to identify all-season roads in Indonesia.

The intersection between the road map with a 2-km radius and all-season road data produced the all-season road map. In the next step, we overlaid this map with a population layer, removed urban areas, and counted the population in the buffer (World Bank 2016). This resulted in the total rural population living within 2 km of all-season roads. Finally, the ratio between the total rural population living within 2 km of all-season roads and the total rural population resulted in the RAI.

The next step was to examine the impact of the Nawacita programme, which is part of the second objective. This study employed the variance coefficient (Equation (1)), the Gini coefficient (Equation (2)), the Lorenz curve, and the Theil index (Equation (3)) to measure rural road inequality. These methods are frequently used to quantify inequity in the transportation sector (e.g., Jang et al. 2017; Mestre 2021; Simon and Natarajan 2017; Zimm 2019). By analysing the inequality values of these different approaches, we can understand how well Indonesia's rural roads are being constructed. In addition, this study also used the decomposition of the inequality indicator (Equation (4)) and convergence analysis (Equations (5) and (6)) to evaluate the implementation of the Nawacita programme.

$$CV_t = \frac{se(RAI_t)}{\overline{RAI}_t} \text{ where } se(RAI_t) = \sqrt{\frac{\sum_{i=1}^{n}\left(RAI_{it} - \overline{RAI}_t\right)^2}{n}} \tag{1}$$

$$Gini_t = 1 - \sum_{i=1}^{n}\left(X_{it} - X_{(i-1)t}\right)\left(Y_{it} + Y_{(i-1)t}\right) \tag{2}$$

$$T_t = \sum_{i=1}^{n} \frac{1}{n}\frac{RAI_{it}}{\overline{RAI}_t} ln \frac{RAI_{it}}{\overline{RAI}_t} \tag{3}$$

$\overline{RAI}_t, se(RAI_t), CV_t, Gini_t$ and $T_t$ are the mean, standard deviation, coefficient of variance, Gini coefficient and Theil index year $t$, respectively. $X_{it}$ is the cumulative proportion of the population variable in the smaller region $i = 1, \ldots, n$ year $t$ with $X_{0t} = 0$ and $X_{nt} = 1$. $Y_{it}$ is the cumulative proportion of the RAI variable in the smaller region $i = 1, \ldots, n$ year $t$ with



$Y_{0t} = 0$ and $Y_{nt} = 1$. $Y_{it}$ should be indexed in non-decreasing order ($Y_{it} \geq Y_{(i-1)t}$) and $X_{it}$ is generated by arranging regions in ascending order based on the RAI values. A lower variation coefficient value indicates a more equitable distribution.

The Gini coefficient is a simple mathematical metric representing the overall degree of inequality, whereas the Lorenz curve is a visual representation of equality. The Gini coefficient is usually calculated from the Lorenz curve. The Gini coefficient is the ratio of the segment between the 45° line of equality and the Lorenz curve over the entire segment under the 45° line. It has a value from 0 to 1, where 0 stands for perfect equality and 1 denotes perfect inequality. The higher the Gini coefficient, the further away the Lorenz curve is from the 45° line. The Lorenz curve is a valuable and essential visualisation tool because different Lorenz curves can have the same Gini coefficient (Zimm 2019). A Gini value of less than 0.20 stands for low inequality, a value from 0.20 to 0.50 shows medium inequality, and a value above 0.50 indicates high inequality.

The Theil index is part of a larger family of measures referred to as the general entropy class. If the Gini coefficient computes the deviation, the Theil index describes the entropic distance between a situation and the ideal egalitarian situation (Mestre 2021). Like the Gini coefficient, the Theil index also ranges from 0 to 1, where 0 stands for perfect equality and 1 denotes perfect inequality.

The decomposition of the inequality indicator assesses the contribution of within-inequality, between-inequality, and a residual term to total inequality (Bellu and Liberati 2006), as shown in Equation (4). Within-inequality captures disparity due to the variability of the RAI within each regional group. Between-inequality shows disparity due to the variability of the RAI across different regional groups. The coefficient of variance and the Gini index are not perfectly decomposable (Bellu and Liberati 2006; Cowell 2011), hence only the Theil index was decomposed. Let us assume that there are $m$ regional groups. The Theil index can be decomposed as follows:

$$T_t = \sum_{k=1}^{m} \frac{n_k}{n} \frac{\overline{RAI_{kt}}}{\overline{RAI_t}} T(RAI_{kt}) + T\left(\overline{RAI_t}\right) \tag{4}$$

$n_k$ is the number of smaller regions in the regional group $k$. $T(RAI_{kt})$ is the Theil index of regional group $k$ in year $t$. $T\left(\overline{RAI_t}\right)$ is calculated by replacing each actual RAI of the regional group with the corresponding means, then computing the Theil index of this fictitious RAI distribution (Bellu and Liberati 2006).

We also checked whether the convergence of the RAI occurred. Convergence measurements can use $\sigma$ convergence (Equation (5)) and $\beta$ convergence (Equation (6)). Because $\sigma$ convergence cannot indicate the significance of convergence itself, this study also used $\beta$ convergence. $\sigma$ convergence refers to the decline in the cross-sectional dispersion (disparity) of a rural road access indicator across regions, that is, whether $\sigma \ convergence_{t+T} < \sigma \ convergence_t$.

The concepts of $\sigma$ and $\beta$ convergences are related. Intuitively, we can see that if the RAI levels of 2 regions become more similar over time, it must be the case that the poor region is growing faster. As an illustration, the RAI in region A starts out being higher than the RAI in region B. There is an initial distance or dispersion between the 2 levels of the RAI. If the growth rate of the RAI in region A is smaller than the growth rate of the RAI in region B between times $t$ and $t + T$, we say that there is $\beta$ convergence. Because dispersion at $t + T$ is smaller than at time $t$, we also say that there is $\sigma$ convergence. In other words, $\beta$ convergence is a necessary condition for $\sigma$ convergence (Sala-i-Martin 1996).

$$\sigma \ convergence_t = \sqrt{\frac{1}{n} \sum_{i=1}^{n} \left(ln \ RAI_{it} - ln \ \overline{RAI_t}\right)^2} \tag{5}$$

Suppose that $\beta$ convergence holds for a group of regions $i$, where $i = 1, 2, \ldots, n$, the RAI in region $i$ at time $t$, corresponding perhaps to annual data, can be approximated by:

$$\frac{1}{T} ln \left( \frac{RAI_{i,t+T}}{RAI_{it}} \right) = \alpha - \beta \, ln \, RAI_{it} + u_{it} \tag{6}$$

where $\alpha$ is an intercept and $u_{it}$ is a disturbance term. The annual growth rate of RAI between $t$ and $t + T$ $\left( \frac{1}{T} ln \left( \frac{RAI_{i,t+T}}{RAI_{it}} \right) \right)$ is inversely related to $ln \, RAI$ at time $t$ ($ln \, RAI_{it}$). The negative sign of the coefficient on $ln \, RAI$ exhibits convergence (Sala-i-Martin 1996). On the contrary, the positive sign of this coefficient indicates divergence. Equation (6) assumes that all regions are structurally similar. They have the same steady state and differ only in terms of their initial conditions. It depicts unconditional $\beta$ convergence (Tselios 2009).

## 3. Results and Discussion

### 3.1. Best Approach for Calculating the RAI

We calculated the RAI for a selection of districts in 2018 using various population distribution data, such as WorldPop and LandScan population distribution data. Because of the absence of regional road quality data, this study utilised the national road network map from the Directorate General of Highways and the accessibility data from the Directorate General of Highways and BPS-Statistics Indonesia. The results, displayed in Table 2, show similar values. The Indonesian RAI ranged from 18.94 per cent to 25 per cent. According to WorldPop data, the proportion of the Indonesian rural population in 2018 was 60.61 per cent. In the same year, LandScan data showed that the percentage of the Indonesian rural population was 64.75 per cent. The RAI using LandScan is higher than the RAI using WorldPop data, whichever RAI methods are used, because the WorldPop dataset has the lowest concentration of population in rural areas. This result is in line with Mikou et al. (2019). In general, with the same method, RAIs using different population distribution datasets have the same pattern, as shown in Figures A2 and A3. Table 2 also displays the Pearson correlations of the RAI between different population distribution datasets for each method over 0.8.

**Table 2.** 2018 Indonesian RAI by road network condition data and population distribution data.

| Method | Road Network Condition Data | Indonesian RAI (per cent) | | Pearson Correlation |
| --- | --- | --- | --- | --- |
| | | **WorldPop** | **LandScan** | |
| 1 | IRI | 18.94 | 21.67 | 0.8732 |
| 2 | Road condition | 21.21 | 24.17 | 0.8790 |
| 3 | Podes | 21.59 | 25.00 | 0.8747 |

Source: Author's calculation.

WorldPop data were chosen for the population layer because the computational process underlying the WorldPop data is fully transparent (Stevens et al. 2015), and the model is considered to be the most accurate and robust among the currently available datasets (World Bank 2016). From three methods using WorldPop data, the descriptive statistics of RAI at the district level were similar. The RAI using IRI, road condition, and Podes data had means of 23.41 per cent, 25.21 per cent, and 25.71 per cent, respectively. These data are also in line with the scatter plots in Figure 2. The Pearson correlation between RAI using Podes data and RAI using IRI data was 0.9475. Furthermore, the correlation between RAI using Podes data and RAI using road condition data was also positive, with a Pearson correlation coefficient value of 0.9833. A one-way analysis of variance (ANOVA) was also used to assess whether there were differences between the three methods. The results concluded that there were no differences between the group means (F (2,1322) = 2.01, $p$ = 0.135)[1].

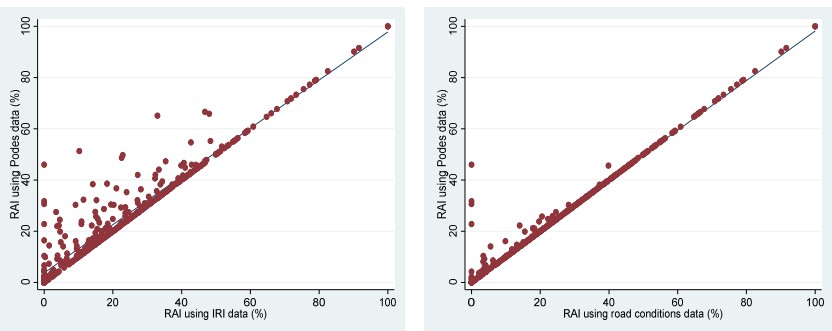

**Figure 2.** Scatter plots between RAI using Podes-WorldPop data and RAI using other methods-WorldPop data in 2018. Source: Author's calculation.

Provincial road quality data from the Directorate General of Highways was unavailable. Based on previous results, this study used population distribution maps from World-Pop, the national and provincial road maps from the Directorate General of Highways, and road network condition data from BPS-Statistics Indonesia to calculate the RAI in 2014, 2018, 2019, and 2020. Table A1 shows the results.

### 3.2. Road Infrastructure Access across Districts in Rural Indonesia

For analysis, this study divided Indonesia into seven regional groups[2]. Figure A4 shows the district locations in each regional group. The results in Figure 3 show that in 2020, rural residents in 3.31 per cent of districts did not live within a two-kilometre radius of all-season national and provincial roads. This data was lower than the 8.56 per cent recorded in 2014. The RAI median also increased from 29.43 per cent in 2014 to 33.68 per cent in 2020. The paired t-test results reached the same conclusion. The 2020 RAI was significantly higher than the 2014 RAI, with a *p*-value of less than 0.001.

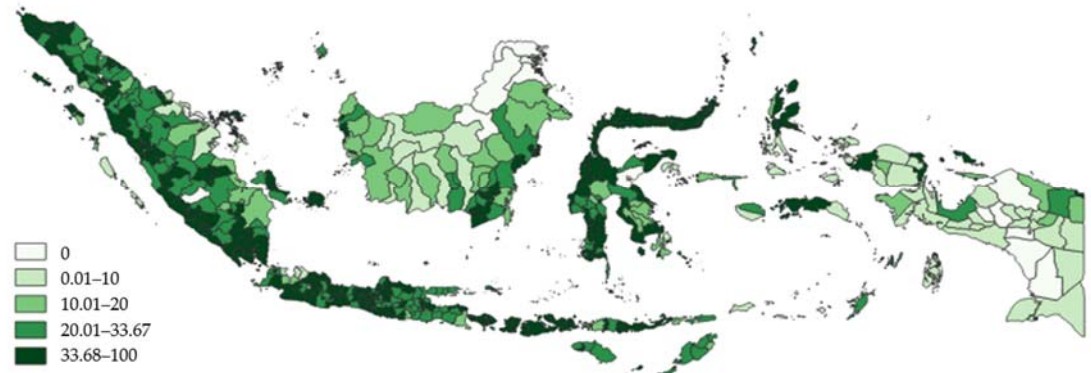

**Figure 3.** 2020 Indonesian RAI by district (per cent). Source: Author's calculation.

The majority of districts with a high RAI are located in four regional groups: Sumatra, Java, Bali and Nusa Tenggara, and Sulawesi. RAI was low in most districts in Borneo, Moluccas, North Moluccas, Papua, and West Papua. During the same time period, 77.38 per cent of districts had a higher RAI. The positive change in RAI occurred in districts with a low RAI, namely in eastern Indonesia, which is the priority of the Nawacita programme (see Figure 4). This shows that the Nawacita programme implementation was relatively successful.

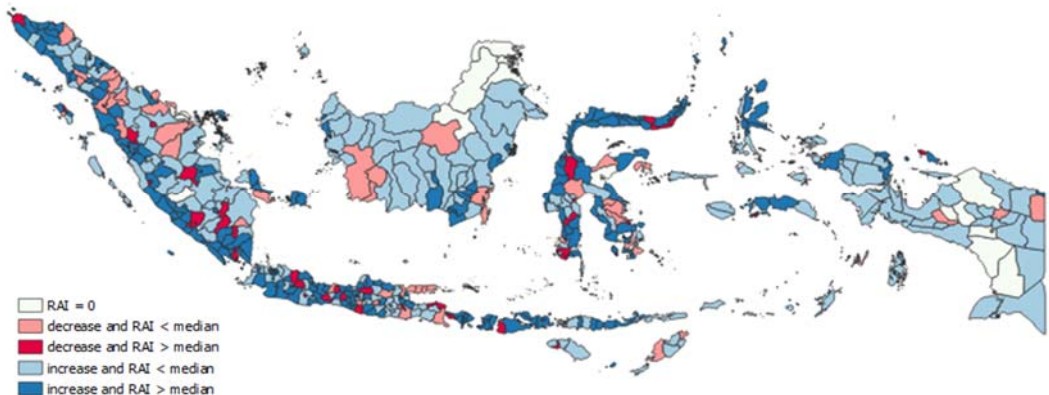

**Figure 4.** Change in the RAI during 2014–2020 by district. Source: Author's calculation.

*3.3. Road Infrastructure Access Inequality in Rural Indonesia*

This study uses the following indicators of inequality to establish the evolution of road infrastructure access inequality in rural Indonesia for 2014–2020: the coefficient of variance, the Gini coefficient, and the Theil index. We decomposed the inequality indicator by region subgroups, using the decomposition technique of Bellu and Liberati (2006); Cowell (2011) and Haughton and Khandker (2009) to analyse the contributions of each region's disparity to total inequality.

The RAI in all regional groups increased significantly after the Nawacita programme's implementation. Bali and Nusa Tenggara had the highest RAI, which increased from 37.62 per cent in 2014 to 44.99 per cent in 2020. Papua and West Papua had the lowest RAI, which reached 9.23 per cent in 2014 and increased to 10.23 per cent in 2020. The policy had a positive impact, reducing Indonesia's inequality between 2014 and 2020. As described in Table 3, the coefficient of variance decreased from 0.665 to 0.587, the Gini coefficient decreased from 0.37 to 0.325, and the Theil index went down from 0.164 to 0.16. Indonesia's Gini coefficient was categorised as "medium inequality". Figure 5 represents the shifts in the Lorenz curve from 2014 to 2020. The results also indicate that inequality fell between 2014 and 2020.

Since 2014, as shown in Table 3, all indicators have demonstrated a consistent declining trend across all Indonesian regions. Java had the lowest level of inequality, while Papua and West Papua had the greatest. Even though Papua and West Papua's rural regions had the lowest RAI and the greatest inequality, this value had decreased. This trend is stronger in this region than in the others.

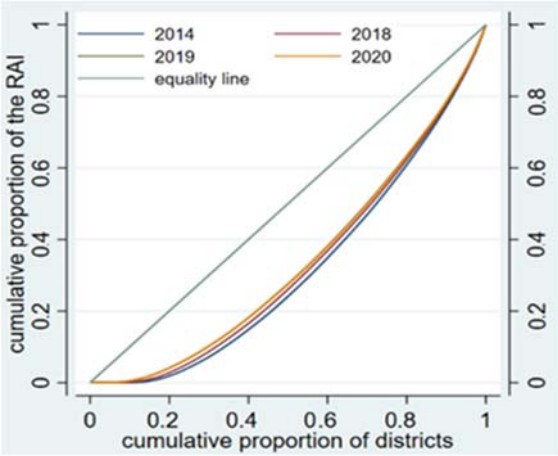

**Figure 5.** Lorenz curve of the RAI. Source: Author's calculation.

**Table 3.** Inequality indicators in 2014–2020 by regional group.

| Inequality Indicators | 2014 | 2018 | 2019 | 2020 |
|---|---|---|---|---|
| Coefficient of variance | | | | |
|   Indonesia | 0.665 | 0.623 | 0.584 | 0.587 |
|   Sumatra | 0.529 | 0.510 | 0.471 | 0.477 |
|   Java | 0.480 | 0.450 | 0.418 | 0.418 |
|   Bali and Nusa Tenggara | 0.531 | 0.452 | 0.447 | 0.450 |
|   Borneo | 0.911 | 0.862 | 0.816 | 0.823 |
|   Sulawesi | 0.569 | 0.543 | 0.502 | 0.503 |
|   Moluccas and North Moluccas | 0.908 | 0.711 | 0.584 | 0.586 |
|   Papua and West Papua | 1.818 | 1.597 | 1.345 | 1.346 |
| Gini coefficient | | | | |
|   Indonesia | 0.370 | 0.345 | 0.324 | 0.325 |
|   Sumatra | 0.285 | 0.273 | 0.254 | 0.256 |
|   Java | 0.250 | 0.235 | 0.219 | 0.218 |
|   Bali and Nusa Tenggara | 0.295 | 0.247 | 0.247 | 0.249 |
|   Borneo | 0.473 | 0.458 | 0.438 | 0.439 |
|   Sulawesi | 0.317 | 0.299 | 0.277 | 0.277 |
|   Moluccas and North Moluccas | 0.484 | 0.393 | 0.324 | 0.326 |
|   Papua and West Papua | 0.769 | 0.714 | 0.657 | 0.656 |
| Theil index | | | | |
|   Indonesia | 0.164 | 0.165 | 0.159 | 0.160 |
|   Sumatra | 0.104 | 0.112 | 0.101 | 0.103 |
|   Java | 0.101 | 0.089 | 0.089 | 0.089 |
|   Bali and Nusa Tenggara | 0.121 | 0.082 | 0.096 | 0.097 |
|   Borneo | 0.281 | 0.257 | 0.228 | 0.230 |
|   Sulawesi | 0.105 | 0.098 | 0.104 | 0.104 |
|   Moluccas and North Moluccas | 0.276 | 0.235 | 0.190 | 0.191 |
|   Papua and West Papua | 0.653 | 0.716 | 0.624 | 0.621 |
| Theil index decomposition | | | | |
|   Within-region inequality | 84.98 | 82.94 | 80.9 | 80.77 |
|   Between-region inequality | 15.02 | 17.06 | 19.1 | 19.23 |

Source: Author's calculation.

The Theil index can be broken down into within-regional and between-region RAI inequalities. In 2020, for instance, we can deduce that Indonesia's inequality was primarily driven (80.77 per cent) by within-regional inequality. In contrast, between-region inequality made a lower contribution to overall inequality at 19.23 per cent. The contribution of within-region inequality has decreased consistently. This trend indicates that the inequality reduction in Indonesia since 2014 has been uniform across geographical locations, whereas the gap between regional groups has risen slightly in recent years.

### 3.4. Convergence of Road Infrastructure Access across Indonesian Districts

Our district analysis captured a strong pattern of regional convergence. As shown in Figure 6, the $\sigma$ convergence of the RAI decreased over time. In 2014, this value was 1.054, and it reached 0.975 in 2020. Table 4 describes the equation of $\beta$ convergence. The regression of the change in the RAI as a function of its initial level confirms the $\beta$ convergence in which the coefficient of the initial value is negative and statistically significant at the 1 per cent level. This means that the rate of increase in the RAI was faster in the district with an initially low RAI and vice versa. The negative trend of $\sigma$ convergence and the negative coefficient of the initial value in the equation of $\beta$ convergence reinforce the previous statement that the Nawacita programme implementation reduced regional inequality during 2014–2020.

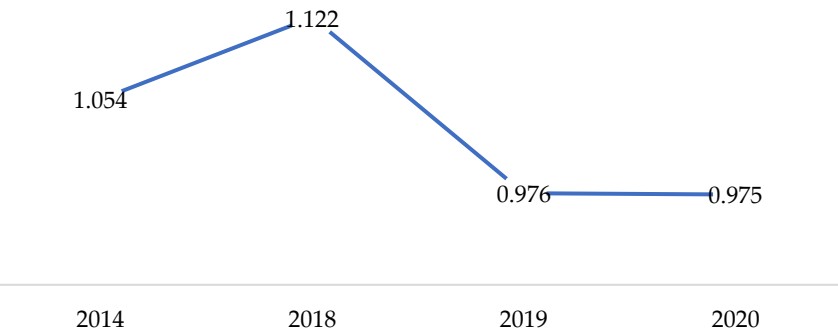

**Figure 6.** The $\sigma$ convergence of RAI across Indonesian districts in 2014–2020. Source: Author's calculation.

**Table 4.** $\beta$ convergence of RAI across Indonesian districts in 2014–2020.

| Dependent Variable: $\frac{1}{6}ln\left(\widehat{\frac{RAI_{i,2020}}{RAI_{i,2014}}}\right)$ | Coefficient | Std. Error | t Statistic | Prob. |
|---|---|---|---|---|
| $ln\ RAI_{i,2014}$ | −0.0599 *** | 0.003 | −17.36 | 0.000 |
| Constant | 0.2255 *** | 0.012 | 19.41 | 0.000 |
| N | 429 | | | |
| R-squared | 0.4138 | | | |
| F-statistic | 301.48 *** | | | |

Note: *** significant at 1 per cent. Source: Author's calculation.

### 3.5. Discussion

From the RAI formula, the change in total rural populations and the change in total rural populations who live within 2 km of all-season roads may drive the inequality reduction and convergence phenomenon of the RAI between districts. Table 5 shows that the median annual growth rate in total rural populations who lived within 2 km of all-season roads between 2014 and 2018 was faster than the median annual growth rate in total rural populations, especially in Papua and West Papua. Based on data from BPS-Statistics Indonesia, the government built 24,557 km of roads between 2014 and 2018, including the Trans-Sumatra, Trans-Borneo, Trans-Sulawesi, Trans-Moluccas, and Trans-Papua roads. This road construction facilitated rural populations' access to all-season roads so that the proportion living within 2 km of all-season roads increased, and improved RAI scores. This argument fits with the values in Table 6 for the Pearson correlations between the RAI and its individual parts. The RAI is strongly linked to rural populations who live within 2 km of all-season roads.

**Table 5.** Median annual growth rate in rural populations and rural populations within 2 km of all-season roads at the district level by regional group (per cent).

| Regional Group | Rural Population | | | Rural Population within 2 km from All-Season Roads | | |
|---|---|---|---|---|---|---|
| | **2014–2018** | **2019** | **2020** | **2014–2018** | **2019** | **2020** |
| Sumatra | 1.27 | 1.36 | 1.42 | 2.55 | 3.32 | 0.35 |
| Java | 0.79 | 0.74 | 0.32 | 1.68 | 0.25 | −0.04 |
| Bali and Nusa Tenggara | 2.04 | 1.88 | 1.94 | 3.45 | 0.77 | 1.43 |
| Borneo | 2.47 | 2.55 | 2.68 | 4.90 | 6.18 | 0.99 |
| Sulawesi | 2.38 | 2.02 | 1.98 | 4.06 | 2.33 | 1.90 |
| Moluccas and North Moluccas | 3.90 | 3.56 | 3.38 | 15.22 | 9.97 | 1.52 |
| Papua and West Papua | 9.57 | 7.87 | 7.46 | 14.53 | 6.30 | 7.30 |
| Indonesia | 2.02 | 1.66 | 1.75 | 3.29 | 2.53 | 0.85 |

Source: Author's calculation.

**Table 6.** Pearson correlation between the RAI and constituent variables.

| Constituent Variable | 2014 | 2018 | 2019 | 2020 |
|---|---|---|---|---|
| Rural populations who live within 2 km of all-season roads | 0.357 *** | 0.3181 *** | 0.2957 *** | 0.2954 *** |
| Rural populations | 0.0464 | −0.0027 | −0.0372 | −0.0458 |

Note: *** significant at 1 per cent. Source: Author's calculation.

Besides road construction, the government can boost the RAI by improving the quality of rural roads. We can use the step-by-step procedures in Figure 1 to calculate the RAI by assuming that the government repairs all existing rural roads so that all rural roads are equal to all-season roads. As shown in Figure 7, the results demonstrate that the increase in all rural road quality did not significantly increase the RAI. The median RAI in Papua and West Papua was still the lowest. Table 7 shows that indicators of inequality decreased slowly. For example, in 2020, the coefficient of variance, Gini coefficient, and Theil index only decreased by 0.051, 0.028 and 0.019, respectively.

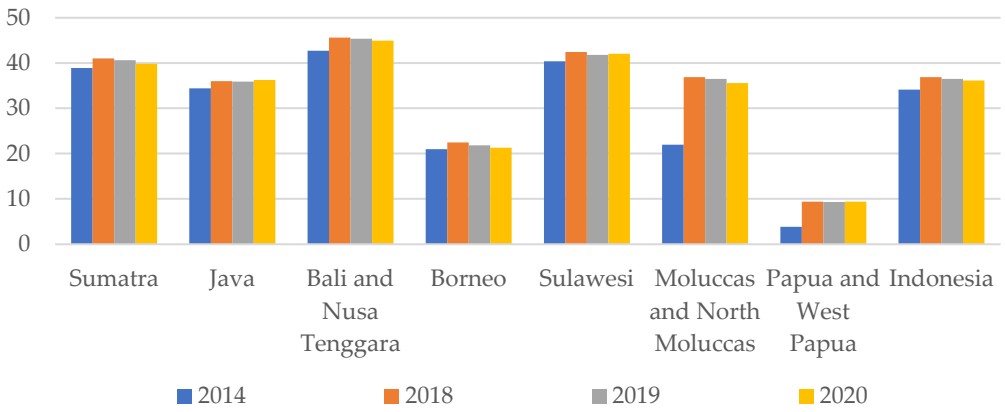

**Figure 7.** The median of the RAI when all rural roads are all-season roads. Source: Author's calculation.

**Table 7.** Real condition and simulation of inequality indicators in 2014–2020.

| Inequality Indicators | 2014 | 2018 | 2019 | 2020 |
|---|---|---|---|---|
| Coefficient of variance | | | | |
| Before | 0.665 | 0.623 | 0.584 | 0.587 |
| After | 0.580 | 0.528 | 0.533 | 0.536 |
| Gini coefficient | | | | |
| Before | 0.370 | 0.345 | 0.324 | 0.325 |
| After | 0.324 | 0.294 | 0.295 | 0.297 |
| Theil index | | | | |
| Before | 0.164 | 0.165 | 0.159 | 0.160 |
| After | 0.153 | 0.137 | 0.139 | 0.141 |

Notes: "Before" shows the real condition. "After" shows the simulation when all rural roads are all-season roads. Source: Author's calculation.

Analysing the link between the RAI and the District Fiscal Capacity Index (DFCI)[3] can help the government decide on the policy priority: new road construction or old road maintenance. For example, as shown in Figure 8, in 2020, the number of districts with low RAI and low DFCI in Moluccas, North Moluccas, Papua, and West Papua was higher than the number of districts in other regional groups. This indicates that the government needs to prioritise the construction of new national and provincial roads in these areas because the district's ability to fund local road development is low. In general, the construction of national and provincial roads is right on target because it is carried out in districts with a low DFCI. However, the construction of national and provincial roads in Bali, Nusa Tenggara, and Borneo requires collaboration between the central, provincial, and district

governments because, financially, the fiscal capacity of districts in these regional groups is relatively good. Coordination can prevent road construction from being concentrated in certain areas and guarantee connectivity between national and regional roads.

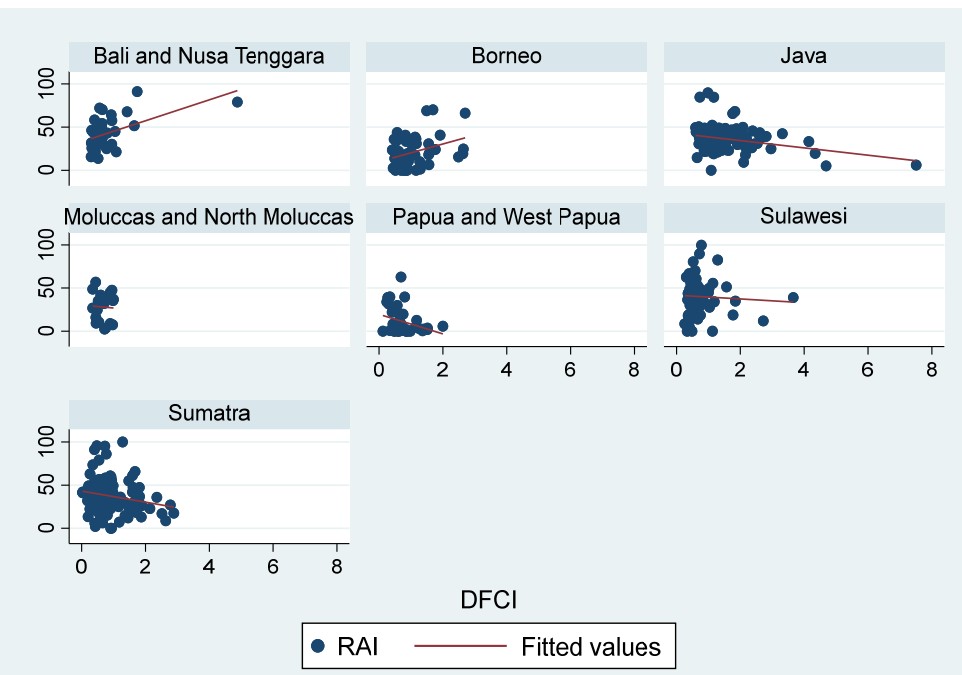

**Figure 8.** Scatter plots between the RAI and DFCI by regional group. Source: Author's calculation.

## 4. Conclusions

The RAI, as SDG indicator 9.1.1, is a relatively good predictor of rural road quality. Due to data limitations, it is challenging to calculate this predictor at the regional level. This study attempted to determine the RAI for each district in Indonesia during 2014–2020. The results show that since the implementation of the Nawacita programme, the RAI has increased, inequality has declined, and there has been a strong pattern of regional convergence. To ensure the future success of this implementation, the government can create regulations to designate several road infrastructure projects as a national strategy. This regulation can specify the types of permits and non-permits that can be expedited by a minister, head of a national agency, or mayor of a region, as well as spatial planning compliance, land availability, and procurement methods. Since most road infrastructure spending comes from the government budget, there needs to be more long-term work to increase tax collection, such as the tax amnesty programme. To encourage more public–private partnerships, the government can also use fiscal policies, such as government guarantees for direct loans.

**Author Contributions:** Conceptualization, R.N.T.W., M.I., A.D. and K.K.; methodology, R.N.T.W.; software, R.N.T.W.; validation, M.I., A.D. and K.K.; formal analysis, R.N.T.W.; investigation, M.I., A.D. and K.K.; resources, R.N.T.W.; data curation, R.N.T.W.; writing—original draft preparation, R.N.T.W.; writing—review and editing, R.N.T.W., M.I., A.D. and K.K.; visualization, R.N.T.W.; supervision, M.I., A.D. and K.K.; project administration, R.N.T.W.; funding acquisition, R.N.T.W. All authors have read and agreed to the published version of the manuscript.

**Funding:** This research received no external funding.

**Informed Consent Statement:** Not applicable.

**Data Availability Statement:** Not applicable.

**Conflicts of Interest:** The author declares no conflict of interest.

## Appendix A

**Table A1.** Indonesian RAI using the national road network map, WorldPop, and Podes by district (per cent).

| Code | District | 2014 | 2018 | 2019 | 2020 | Code | District | 2014 | 2018 | 2019 | 2020 |
|------|----------|------|------|------|------|------|----------|------|------|------|------|
| 1101 | Simeulue | 32.3 | 37.8 | 45.5 | 45.0 | 3672 | Cilegon | 0.0 | 0.0 | 0.0 | 0.0 |
| 1102 | Aceh Singkil | 13.2 | 12.9 | 16.0 | 15.8 | 3673 | Serang | 35.6 | 36.5 | 37.2 | 37.5 |
| 1103 | Aceh Selatan | 36.1 | 37.9 | 42.4 | 41.6 | 5101 | Jembrana | 57.9 | 58.5 | 57.7 | 57.6 |
| 1104 | Aceh Tenggara | 28.0 | 29.0 | 29.0 | 28.5 | 5102 | Tabanan | 69.5 | 70.0 | 70.0 | 70.2 |
| 1105 | Aceh Timur | 33.9 | 35.5 | 33.0 | 32.0 | 5103 | Badung | 78.7 | 78.9 | 78.9 | 79.0 |
| 1106 | Aceh Tengah | 39.8 | 46.2 | 47.0 | 46.2 | 5104 | Gianyar | 86.5 | 91.4 | 91.4 | 91.2 |
| 1107 | Aceh Barat | 27.4 | 28.2 | 30.5 | 29.8 | 5105 | Klungkung | 24.6 | 24.5 | 24.1 | 25.0 |
| 1108 | Aceh Besar | 64.6 | 65.6 | 61.6 | 61.1 | 5106 | Bangli | 65.3 | 66.6 | 72.3 | 72.1 |
| 1109 | Pidie | 42.4 | 43.9 | 44.2 | 43.1 | 5107 | Karang Asem | 67.3 | 67.3 | 66.7 | 67.3 |
| 1110 | Bireuen | 42.3 | 44.7 | 55.0 | 55.0 | 5108 | Buleleng | 67.5 | 68.0 | 67.7 | 67.8 |
| 1111 | Aceh Utara | 33.7 | 34.8 | 37.5 | 36.8 | 5201 | Lombok Barat | 39.7 | 40.5 | 64.2 | 64.3 |
| 1112 | Aceh Barat Daya | 46.2 | 47.9 | 55.7 | 55.5 | 5202 | Lombok Tengah | 28.4 | 39.2 | 44.8 | 44.9 |
| 1113 | Gayo Lues | 30.6 | 32.3 | 32.0 | 31.0 | 5203 | Lombok Timur | 30.4 | 30.0 | 51.6 | 51.6 |
| 1114 | Aceh Tamiang | 13.3 | 14.1 | 14.3 | 13.5 | 5204 | Sumbawa | 40.1 | 43.0 | 43.3 | 43.2 |
| 1115 | Nagan Raya | 34.9 | 43.7 | 46.2 | 45.4 | 5205 | Dompu | 52.7 | 54.6 | 54.3 | 54.2 |
| 1116 | Aceh Jaya | 33.1 | 30.6 | 35.2 | 34.3 | 5206 | Bima | 46.3 | 48.5 | 49.5 | 49.3 |
| 1117 | Bener Meriah | 24.3 | 34.8 | 41.6 | 41.2 | 5207 | Sumbawa Barat | 37.9 | 37.8 | 37.3 | 37.3 |
| 1118 | Pidie Jaya | 55.2 | 57.7 | 56.2 | 55.5 | 5208 | Lombok Utara | 44.6 | 48.5 | 50.7 | 50.3 |
| 1172 | Sabang | 59.4 | 91.2 | 91.2 | 91.1 | 5272 | Bima | 57.3 | 58.7 | 57.8 | 58.3 |
| 1173 | Langsa | 41.1 | 39.8 | 72.7 | 73.5 | 5301 | Sumba Barat | 52.0 | 55.1 | 47.8 | 46.3 |
| 1174 | Lhokseumawe | 57.5 | 53.9 | 52.5 | 52.6 | 5302 | Sumba Timur | 30.5 | 31.2 | 32.2 | 32.6 |
| 1175 | Subulussalam | 43.8 | 45.3 | 45.6 | 45.0 | 5303 | Kupang | 30.3 | 31.7 | 30.5 | 29.6 |
| 1201 | Nias | 25.2 | 31.0 | 39.5 | 38.9 | 5304 | Timor Tengah Selatan | 18.3 | 18.5 | 21.6 | 21.4 |
| 1202 | Mandailing Natal | 36.8 | 38.5 | 39.7 | 39.0 | 5305 | Timor Tengah Utara | 29.1 | 31.7 | 31.3 | 31.1 |
| 1203 | Tapanuli Selatan | 34.3 | 34.5 | 33.6 | 33.5 | 5306 | Belu | 31.3 | 32.3 | 30.4 | 30.1 |
| 1204 | Tapanuli Tengah | 49.4 | 51.3 | 50.3 | 50.1 | 5307 | Alor | 17.1 | 19.7 | 22.5 | 21.9 |
| 1205 | Tapanuli Utara | 40.2 | 42.9 | 43.5 | 43.3 | 5308 | Lembata | 11.5 | 21.4 | 15.9 | 15.7 |
| 1206 | Toba Samosir | 44.6 | 44.6 | 47.0 | 45.8 | 5309 | Flores Timur | 34.2 | 40.3 | 42.9 | 43.2 |
| 1207 | Labuhan Batu | 24.7 | 25.5 | 28.6 | 28.1 | 5310 | Sikka | 38.6 | 39.9 | 40.5 | 40.1 |
| 1208 | Asahan | 26.7 | 28.1 | 27.0 | 26.0 | 5311 | Ende | 43.5 | 44.6 | 43.8 | 44.4 |
| 1209 | Simalungun | 30.6 | 31.7 | 32.8 | 32.0 | 5312 | Ngada | 38.8 | 39.9 | 39.3 | 39.2 |
| 1210 | Dairi | 29.5 | 31.0 | 28.8 | 28.4 | 5313 | Manggarai | 37.5 | 40.2 | 38.6 | 38.0 |
| 1211 | Karo | 41.8 | 42.7 | 43.9 | 43.2 | 5314 | Rote Ndao | 22.2 | 33.4 | 31.8 | 31.7 |
| 1212 | Deli Serdang | 25.4 | 26.5 | 29.2 | 29.3 | 5315 | Manggarai Barat | 12.9 | 15.0 | 14.6 | 13.7 |
| 1213 | Langkat | 20.8 | 25.1 | 25.7 | 25.0 | 5316 | Sumba Tengah | 22.2 | 22.9 | 25.0 | 24.4 |
| 1214 | Nias Selatan | 16.6 | 17.8 | 19.5 | 19.2 | 5317 | Sumba Barat Daya | 22.4 | 23.0 | 26.0 | 25.7 |
| 1215 | Humbang Hasundutan | 26.6 | 26.5 | 26.6 | 26.6 | 5318 | Nagekeo | 45.5 | 45.6 | 45.5 | 46.1 |
| 1216 | Pakpak Bharat | 30.3 | 29.9 | 30.3 | 30.0 | 5319 | Manggarai Timur | 23.9 | 25.8 | 28.5 | 28.2 |
| 1217 | Samosir | 30.7 | 33.3 | 46.3 | 45.8 | 5320 | Sabu Raijua | 14.6 | 47.2 | 39.8 | 40.4 |
| 1218 | Serdang Bedagai | 36.9 | 36.9 | 38.7 | 37.7 | 5321 | Malaka | 0.0 | 0.0 | 15.9 | 15.9 |
| 1219 | Batu Bara | 32.1 | 32.1 | 34.2 | 34.4 | 5371 | Kota Kupang | 8.5 | 40.1 | 92.9 | 92.9 |
| 1220 | Padang Lawas Utara | 32.4 | 33.4 | 31.6 | 30.7 | 6101 | Sambas | 11.2 | 18.7 | 20.2 | 19.5 |
| 1221 | Padang Lawas | 34.5 | 35.5 | 39.9 | 39.8 | 6102 | Bengkayang | 10.6 | 25.0 | 23.9 | 23.5 |
| 1222 | Labuhan Batu Selatan | 22.8 | 23.9 | 22.4 | 22.0 | 6103 | Landak | 14.9 | 16.0 | 16.4 | 15.8 |
| 1223 | Labuhan Batu Utara | 17.8 | 17.8 | 16.9 | 17.0 | 6104 | Pontianak | 41.8 | 44.2 | 48.6 | 47.8 |
| 1224 | Nias Utara | 33.0 | 33.5 | 38.9 | 38.3 | 6105 | Sanggau | 16.7 | 18.7 | 18.3 | 17.8 |
| 1225 | Nias Barat | 28.3 | 29.1 | 20.6 | 20.3 | 6106 | Ketapang | 11.3 | 11.7 | 10.5 | 10.5 |
| 1276 | Binjai | 19.9 | 20.3 | 19.8 | 20.2 | 6107 | Sintang | 6.0 | 6.3 | 6.6 | 6.5 |
| 1277 | Padangsidimpuan | 72.5 | 72.3 | 72.4 | 72.4 | 6108 | Kapuas Hulu | 13.4 | 13.8 | 14.5 | 14.5 |
| 1278 | Gunungsitoli | 64.4 | 66.0 | 64.8 | 64.3 | 6109 | Sekadau | 8.9 | 10.8 | 10.3 | 9.9 |
| 1301 | Kepulauan Mentawai | 0.0 | 7.6 | 6.0 | 5.8 | 6110 | Melawi | 2.5 | 2.6 | 4.4 | 4.4 |
| 1302 | Pesisir Selatan | 16.5 | 18.7 | 30.6 | 29.2 | 6111 | Kayong Utara | 19.3 | 21.3 | 22.2 | 21.9 |
| 1303 | Solok | 39.9 | 44.0 | 43.4 | 41.9 | 6112 | Kubu Raya | 11.3 | 11.9 | 19.9 | 19.6 |
| 1304 | Sijunjung | 36.2 | 38.7 | 40.6 | 38.6 | 6172 | Singkawang | 36.0 | 38.0 | 37.1 | 36.0 |
| 1305 | Tanah Datar | 58.1 | 59.9 | 59.3 | 58.2 | 6201 | Kotawaringin Barat | 19.0 | 19.8 | 19.8 | 19.7 |
| 1306 | Padang Pariaman | 49.4 | 50.4 | 57.0 | 56.1 | 6202 | Kotawaringin Timur | 14.2 | 14.5 | 19.4 | 19.4 |
| 1307 | Agam | 38.2 | 38.6 | 40.1 | 39.3 | 6203 | Kapuas | 5.0 | 6.5 | 10.2 | 9.9 |

**Table A1.** *Cont.*

| Code | District | 2014 | 2018 | 2019 | 2020 | Code | District | 2014 | 2018 | 2019 | 2020 |
|------|----------|------|------|------|------|------|----------|------|------|------|------|
| 1308 | Lima Puluh Kota | 34.7 | 36.6 | 34.8 | 34.0 | 6204 | Barito Selatan | 9.1 | 12.0 | 12.5 | 12.4 |
| 1309 | Pasaman | 31.4 | 34.5 | 32.4 | 30.6 | 6205 | Barito Utara | 3.2 | 5.7 | 16.8 | 16.7 |
| 1310 | Solok Selatan | 28.1 | 30.8 | 31.3 | 29.5 | 6206 | Sukamara | 2.7 | 2.7 | 2.6 | 2.5 |
| 1311 | Dharmasraya | 24.4 | 25.6 | 25.8 | 25.6 | 6207 | Lamandau | 14.5 | 13.4 | 13.5 | 13.7 |
| 1312 | Pasaman Barat | 36.1 | 37.8 | 40.3 | 39.3 | 6208 | Seruyan | 1.6 | 1.6 | 6.2 | 6.5 |
| 1371 | Padang | 46.0 | 46.6 | 46.7 | 47.3 | 6209 | Katingan | 2.4 | 3.9 | 4.6 | 4.5 |
| 1372 | Solok | 40.3 | 41.0 | 40.1 | 39.8 | 6210 | Pulang Pisau | 28.6 | 33.1 | 34.0 | 33.7 |
| 1373 | Sawah Lunto | 44.6 | 46.3 | 46.7 | 46.7 | 6211 | Gunung Mas | 7.7 | 9.4 | 17.8 | 17.8 |
| 1374 | Padang Panjang | 100.0 | 95.4 | 95.3 | 95.6 | 6212 | Barito Timur | 9.3 | 10.2 | 18.3 | 18.2 |
| 1376 | Payakumbuh | 58.4 | 62.4 | 31.4 | 31.5 | 6213 | Murung Raya | 3.4 | 2.6 | 2.9 | 2.6 |
| 1377 | Pariaman | 67.3 | 78.3 | 78.2 | 79.0 | 6271 | Palangka Raya | 16.7 | 16.5 | 30.9 | 30.6 |
| 1401 | Kuantan Singingi | 23.0 | 25.2 | 26.4 | 25.9 | 6301 | Tanah Laut | 34.9 | 42.0 | 41.6 | 40.7 |
| 1402 | Indragiri Hulu | 28.2 | 24.8 | 26.0 | 26.1 | 6302 | Kota Baru | 20.4 | 22.3 | 19.8 | 19.2 |
| 1403 | Indragiri Hilir | 5.9 | 4.4 | 7.1 | 7.2 | 6303 | Banjar | 32.9 | 36.4 | 39.5 | 38.4 |
| 1404 | Pelalawan | 15.3 | 15.2 | 14.7 | 14.7 | 6304 | Barito Kuala | 27.6 | 29.3 | 33.3 | 32.9 |
| 1405 | Siak | 24.5 | 26.7 | 30.3 | 30.1 | 6305 | Tapin | 25.3 | 29.6 | 31.5 | 30.9 |
| 1406 | Kampar | 24.6 | 32.4 | 32.5 | 31.6 | 6306 | Hulu Sungai Selatan | 43.7 | 45.9 | 44.7 | 43.7 |
| 1407 | Rokan Hulu | 20.3 | 21.6 | 22.4 | 22.7 | 6307 | Hulu Sungai Tengah | 34.3 | 37.3 | 37.7 | 36.7 |
| 1408 | Bengkalis | 18.0 | 17.9 | 8.9 | 8.7 | 6308 | Hulu Sungai Utara | 26.8 | 29.6 | 30.8 | 30.0 |
| 1409 | Rokan Hilir | 25.2 | 27.0 | 26.5 | 26.3 | 6309 | Tabalong | 29.3 | 34.7 | 34.7 | 33.5 |
| 1410 | Kepulauan Meranti | 0.0 | 0.0 | 0.0 | 0.0 | 6310 | Tanah Bumbu | 25.8 | 29.5 | 32.0 | 30.7 |
| 1471 | Pekanbaru | 36.4 | 36.2 | 35.2 | 35.9 | 6311 | Balangan | 33.8 | 37.3 | 38.0 | 36.4 |
| 1473 | Dumai | 23.0 | 23.2 | 61.0 | 60.7 | 6371 | Banjarmasin | 15.4 | 15.9 | 70.1 | 70.1 |
| 1501 | Kerinci | 33.1 | 34.1 | 36.5 | 35.3 | 6372 | Banjar Baru | 77.5 | 78.6 | 78.4 | 78.3 |
| 1502 | Merangin | 26.3 | 25.9 | 27.8 | 28.0 | 6401 | Paser | 22.2 | 24.2 | 24.1 | 24.1 |
| 1503 | Sarolangun | 31.0 | 31.3 | 32.9 | 32.8 | 6402 | Kutai Barat | 16.8 | 17.1 | 19.3 | 19.3 |
| 1504 | Batang Hari | 35.6 | 36.0 | 35.0 | 35.0 | 6403 | Kutai Kartanegara | 23.2 | 24.5 | 24.9 | 24.7 |
| 1505 | Muaro Jambi | 25.1 | 38.0 | 42.5 | 42.0 | 6404 | Kutai Timur | 14.6 | 15.7 | 15.7 | 15.5 |
| 1506 | Tanjung Jabung Timur | 13.6 | 13.2 | 14.4 | 14.5 | 6405 | Berau | 12.9 | 16.8 | 18.3 | 18.0 |
| 1507 | Tanjung Jabung Barat | 15.2 | 16.9 | 29.5 | 29.6 | 6409 | Penajam Paser Utara | 34.4 | 35.6 | 40.5 | 40.7 |
| 1508 | Tebo | 24.3 | 25.7 | 29.1 | 29.4 | 6411 | Mahakam Ulu | 0.0 | 0.0 | 0.0 | 0.0 |
| 1509 | Bungo | 34.4 | 36.2 | 35.8 | 35.2 | 6471 | Balikpapan | 66.1 | 66.5 | 65.8 | 66.2 |
| 1571 | Jambi | 86.9 | 100.0 | 100.0 | 100.0 | 6472 | Samarinda | 9.1 | 9.2 | 68.6 | 69.0 |
| 1572 | Sungai Penuh | 56.0 | 56.8 | 55.6 | 55.6 | 6474 | Bontang | 1.3 | 1.3 | 1.3 | 1.4 |
| 1601 | Ogan Komering Ulu | 24.8 | 26.4 | 26.6 | 25.2 | 6501 | Malinau | 0.0 | 0.0 | 0.0 | 0.0 |
| 1602 | Ogan Komering Ilir | 17.1 | 18.5 | 18.2 | 18.1 | 6502 | Bulungan | 0.0 | 0.0 | 0.0 | 0.0 |
| 1603 | Muara Enim | 25.9 | 27.3 | 26.5 | 26.3 | 6503 | Tana Tidung | 0.0 | 0.0 | 0.0 | 0.0 |
| 1604 | Lahat | 34.5 | 34.9 | 33.9 | 33.7 | 6504 | Nunukan | 0.0 | 0.0 | 0.0 | 0.0 |
| 1605 | Musi Rawas | 21.4 | 22.5 | 22.7 | 22.1 | 6571 | Tarakan | 0.0 | 0.0 | 0.0 | 0.0 |
| 1606 | Musi Banyuasin | 17.1 | 17.7 | 23.4 | 22.9 | 7101 | Bolaang Mongondow | 39.2 | 44.9 | 49.9 | 49.2 |
| 1607 | Banyuasin | 11.0 | 11.2 | 13.6 | 13.1 | 7102 | Minahasa | 45.8 | 46.4 | 70.0 | 70.1 |
| 1608 | Ogan Komering Ulu Selatan | 30.7 | 32.4 | 33.2 | 33.2 | 7103 | Kepulauan Sangihe | 59.8 | 62.4 | 61.9 | 62.2 |
| 1609 | Ogan Komering Ulu Timur | 41.0 | 41.6 | 40.2 | 39.8 | 7104 | Kepulauan Talaud | 53.7 | 56.0 | 65.4 | 66.7 |
| 1610 | Ogan Ilir | 36.2 | 36.6 | 36.5 | 36.1 | 7105 | Minahasa Selatan | 53.9 | 56.2 | 62.4 | 61.7 |
| 1611 | Empat Lawang | 26.9 | 28.9 | 39.9 | 38.9 | 7106 | Minahasa Utara | 59.8 | 61.7 | 61.1 | 60.7 |
| 1612 | Penukal Abab Lematang Ilir | 0.0 | 0.0 | 12.0 | 12.1 | 7107 | Bolaang Mongondow Utara | 57.3 | 56.2 | 56.2 | 55.9 |
| 1613 | Musi Rawas Utara | 0.0 | 0.0 | 0.0 | 0.0 | 7108 | Siau Tagulandang Biaro | 9.6 | 9.5 | 9.2 | 8.9 |
| 1671 | Palembang | 15.9 | 16.1 | 18.0 | 17.5 | 7109 | Minahasa Tenggara | 61.5 | 63.5 | 63.0 | 62.8 |
| 1672 | Prabumulih | 52.9 | 55.6 | 55.6 | 56.7 | 7110 | Bolaang Mongondow Selatan | 46.7 | 52.5 | 44.0 | 43.9 |
| 1673 | Pagar Alam | 51.4 | 52.4 | 51.8 | 51.0 | 7111 | Bolaang Mongondow Timur | 59.9 | 60.6 | 50.6 | 49.9 |
| 1674 | Lubuklinggau | 51.4 | 60.8 | 61.1 | 61.2 | 7171 | Manado | 21.0 | 24.3 | 82.3 | 82.5 |
| 1701 | Bengkulu Selatan | 58.0 | 65.7 | 65.8 | 65.7 | 7172 | Bitung | 38.2 | 38.3 | 38.4 | 39.0 |
| 1702 | Rejang Lebong | 49.2 | 49.9 | 49.4 | 49.4 | 7173 | Tomohon | 80.4 | 80.3 | 80.3 | 80.4 |
| 1703 | Bengkulu Utara | 31.8 | 33.9 | 39.2 | 38.7 | 7174 | Kotamobagu | 59.3 | 89.4 | 89.6 | 89.6 |
| 1704 | Kaur | 49.7 | 47.0 | 51.4 | 53.0 | 7201 | Banggai Kepulauan | 8.8 | 9.0 | 8.8 | 8.5 |

**Table A1.** *Cont.*

| Code | District | 2014 | 2018 | 2019 | 2020 | Code | District | 2014 | 2018 | 2019 | 2020 |
|------|----------|------|------|------|------|------|----------|------|------|------|------|
| 1705 | Seluma | 45.5 | 46.7 | 48.6 | 48.1 | 7202 | Banggai | 35.5 | 38.8 | 48.9 | 48.9 |
| 1706 | Mukomuko | 37.8 | 37.7 | 39.8 | 40.0 | 7203 | Morowali | 30.6 | 30.4 | 39.1 | 40.9 |
| 1707 | Lebong | 41.7 | 42.2 | 42.2 | 41.7 | 7204 | Poso | 42.8 | 41.7 | 46.5 | 46.7 |
| 1708 | Kepahiang | 62.3 | 64.6 | 64.0 | 62.9 | 7205 | Donggala | 41.3 | 42.4 | 41.3 | 41.6 |
| 1709 | Bengkulu Tengah | 45.9 | 54.0 | 49.2 | 49.1 | 7206 | Toli-Toli | 35.8 | 37.0 | 46.5 | 46.5 |
| 1771 | Bengkulu | 34.1 | 36.9 | 36.8 | 36.1 | 7207 | Buol | 41.6 | 44.0 | 46.3 | 46.4 |
| 1801 | Lampung Barat | 24.4 | 25.0 | 43.0 | 43.5 | 7208 | Parigi Moutong | 34.8 | 35.2 | 43.7 | 44.2 |
| 1802 | Tanggamus | 34.8 | 38.8 | 47.9 | 47.9 | 7209 | Tojo Una-Una | 25.6 | 26.6 | 25.1 | 25.1 |
| 1803 | Lampung Selatan | 45.3 | 46.3 | 49.4 | 49.4 | 7210 | Sigi | 50.1 | 50.9 | 49.8 | 49.4 |
| 1804 | Lampung Timur | 44.0 | 44.7 | 47.8 | 47.3 | 7211 | Banggai Laut | 0.0 | 0.0 | 0.0 | 0.0 |
| 1805 | Lampung Tengah | 42.3 | 42.2 | 44.1 | 43.7 | 7212 | Morowali Utara | 0.0 | 0.0 | 0.0 | 0.0 |
| 1806 | Lampung Utara | 46.5 | 48.4 | 47.5 | 47.0 | 7271 | Palu | 6.9 | 8.0 | 11.7 | 11.9 |
| 1807 | Way Kanan | 39.2 | 39.5 | 42.4 | 41.9 | 7301 | Kepulauan Selayar | 29.3 | 34.4 | 33.3 | 32.0 |
| 1808 | Tulangbawang | 28.7 | 28.6 | 33.2 | 33.7 | 7302 | Bulukumba | 41.0 | 42.7 | 50.0 | 49.5 |
| 1809 | Pesawaran | 52.2 | 53.2 | 45.4 | 44.9 | 7303 | Bantaeng | 36.0 | 41.0 | 39.3 | 38.3 |
| 1810 | Pringsewu | 48.5 | 49.9 | 59.8 | 59.7 | 7304 | Jeneponto | 51.8 | 53.0 | 51.5 | 51.2 |
| 1811 | Mesuji | 27.9 | 26.1 | 24.7 | 24.8 | 7305 | Takalar | 28.8 | 29.2 | 29.7 | 29.0 |
| 1812 | Tulang Bawang Barat | 44.2 | 45.2 | 44.5 | 44.1 | 7306 | Gowa | 45.0 | 45.3 | 44.6 | 44.6 |
| 1813 | Pesisir Barat | 0.0 | 0.0 | 42.2 | 42.3 | 7307 | Sinjai | 29.4 | 32.9 | 32.3 | 31.4 |
| 1871 | Bandar Lampung | 16.5 | 17.0 | 17.6 | 17.0 | 7308 | Maros | 31.9 | 32.4 | 32.4 | 32.0 |
| 1872 | Metro | 93.9 | 94.8 | 94.8 | 95.1 | 7309 | Pangkajene Dan Kepulauan | 33.5 | 35.2 | 33.7 | 33.0 |
| 1901 | Bangka | 35.4 | 36.1 | 35.9 | 35.7 | 7310 | Barru | 43.0 | 44.4 | 44.5 | 43.8 |
| 1902 | Belitung | 29.2 | 37.3 | 47.3 | 46.1 | 7311 | Bone | 30.9 | 35.5 | 34.5 | 34.1 |
| 1903 | Bangka Barat | 20.9 | 20.0 | 21.4 | 22.3 | 7312 | Soppeng | 41.2 | 43.5 | 47.4 | 46.4 |
| 1904 | Bangka Tengah | 34.3 | 36.0 | 38.2 | 37.3 | 7313 | Wajo | 38.6 | 39.1 | 39.1 | 39.0 |
| 1905 | Bangka Selatan | 29.1 | 29.4 | 28.9 | 28.7 | 7314 | Sidenreng Rappang | 48.0 | 50.3 | 48.1 | 46.6 |
| 1906 | Belitung Timur | 39.8 | 39.5 | 41.7 | 42.1 | 7315 | Pinrang | 32.6 | 33.0 | 34.8 | 34.8 |
| 1971 | Pangkal Pinang | 0.2 | 0.2 | 85.4 | 86.0 | 7316 | Enrekang | 31.8 | 34.9 | 32.7 | 31.8 |
| 2101 | Karimun | 1.8 | 2.3 | 2.2 | 2.1 | 7317 | Luwu | 24.3 | 27.7 | 28.0 | 27.8 |
| 2102 | Bintan | 43.7 | 55.7 | 55.5 | 55.0 | 7318 | Tana Toraja | 24.3 | 23.4 | 28.5 | 28.0 |
| 2103 | Natuna | 23.0 | 24.0 | 24.2 | 24.2 | 7322 | Luwu Utara | 14.9 | 15.4 | 14.3 | 14.0 |
| 2104 | Lingga | 0.0 | 6.6 | 6.4 | 6.3 | 7325 | Luwu Timur | 22.6 | 22.8 | 24.2 | 25.4 |
| 2105 | Kepulauan Anambas | 0.0 | 8.3 | 15.7 | 15.2 | 7326 | Toraja Utara | 15.2 | 17.8 | 19.6 | 18.6 |
| 2171 | Batam | 0.0 | 27.4 | 27.4 | 27.0 | 7371 | Makassar | 0.0 | 0.0 | 0.0 | 0.0 |
| 2172 | Tanjung Pinang | 5.1 | 51.3 | 48.4 | 47.6 | 7372 | Parepare | 6.9 | 73.3 | 72.6 | 72.4 |
| 3201 | Bogor | 17.7 | 18.0 | 19.5 | 19.6 | 7373 | Palopo | 53.1 | 54.7 | 53.9 | 53.0 |
| 3202 | Sukabumi | 30.8 | 39.0 | 42.0 | 41.2 | 7401 | Buton | 19.8 | 21.1 | 16.4 | 17.2 |
| 3203 | Cianjur | 30.2 | 38.2 | 39.8 | 39.2 | 7402 | Muna | 22.6 | 31.2 | 18.9 | 19.3 |
| 3204 | Bandung | 35.8 | 37.8 | 37.3 | 37.2 | 7403 | Konawe | 19.7 | 19.5 | 18.4 | 18.8 |
| 3205 | Garut | 37.2 | 45.6 | 46.1 | 45.7 | 7404 | Kolaka | 36.4 | 38.5 | 40.9 | 41.4 |
| 3206 | Tasikmalaya | 20.4 | 25.8 | 23.6 | 23.3 | 7405 | Konawe Selatan | 39.8 | 41.1 | 39.6 | 40.1 |
| 3207 | Ciamis | 23.2 | 25.0 | 24.4 | 24.4 | 7406 | Bombana | 29.2 | 33.1 | 33.9 | 33.9 |
| 3208 | Kuningan | 33.2 | 35.0 | 36.5 | 36.3 | 7407 | Wakatobi | 0.0 | 21.3 | 20.8 | 20.6 |
| 3209 | Cirebon | 44.6 | 46.1 | 47.9 | 48.2 | 7408 | Kolaka Utara | 40.8 | 39.8 | 42.9 | 43.8 |
| 3210 | Majalengka | 26.3 | 25.6 | 25.9 | 26.5 | 7409 | Buton Utara | 4.9 | 5.1 | 5.2 | 5.6 |
| 3211 | Sumedang | 44.7 | 45.1 | 44.2 | 44.6 | 7410 | Konawe Utara | 16.8 | 16.9 | 14.7 | 15.1 |
| 3212 | Indramayu | 34.5 | 35.4 | 38.3 | 38.5 | 7411 | Kolaka Timur | 0.0 | 0.0 | 17.4 | 18.2 |
| 3213 | Subang | 34.9 | 34.5 | 34.4 | 34.5 | 7471 | Kendari | 85.6 | 85.7 | 55.9 | 55.5 |
| 3214 | Purwakarta | 35.3 | 34.9 | 36.0 | 36.6 | 7472 | Baubau | 49.3 | 51.0 | 64.0 | 64.1 |
| 3215 | Karawang | 6.6 | 8.4 | 8.9 | 9.3 | 7501 | Boalemo | 34.3 | 35.2 | 35.5 | 36.4 |
| 3216 | Bekasi | 4.7 | 4.9 | 5.1 | 5.1 | 7502 | Gorontalo | 46.9 | 47.8 | 51.4 | 51.3 |
| 3217 | Bandung Barat | 24.7 | 25.4 | 29.8 | 29.8 | 7503 | Pohuwato | 38.9 | 43.6 | 45.8 | 46.3 |
| 3218 | Pangandaran | 0.0 | 0.0 | 50.1 | 49.5 | 7504 | Bone Bolango | 44.0 | 44.1 | 49.0 | 49.5 |
| 3278 | Tasikmalaya | 49.5 | 50.4 | 49.5 | 48.9 | 7505 | Gorontalo Utara | 25.0 | 33.8 | 62.4 | 62.6 |
| 3279 | Banjar | 49.5 | 50.1 | 50.2 | 50.4 | 7571 | Gorontalo | 0.0 | 0.0 | 99.6 | 99.6 |
| 3301 | Cilacap | 42.8 | 47.9 | 47.1 | 46.7 | 7601 | Majene | 17.8 | 18.8 | 48.5 | 49.1 |
| 3302 | Banyumas | 51.9 | 50.7 | 49.9 | 49.8 | 7602 | Polewali Mandar | 29.8 | 34.2 | 31.4 | 30.1 |
| 3303 | Purbalingga | 24.8 | 25.7 | 25.2 | 25.5 | 7603 | Mamasa | 15.6 | 21.6 | 29.3 | 30.1 |
| 3304 | Banjarnegara | 48.2 | 49.4 | 49.0 | 48.6 | 7604 | Mamuju | 30.4 | 33.4 | 34.3 | 34.4 |
| 3305 | Kebumen | 20.9 | 21.2 | 20.9 | 21.1 | 7605 | Mamuju Utara | 33.6 | 35.4 | 36.0 | 35.3 |
| 3306 | Purworejo | 38.0 | 37.4 | 38.5 | 38.7 | 7606 | Mamuju Tengah | 0.0 | 0.0 | 38.4 | 37.8 |
| 3307 | Wonosobo | 51.1 | 52.1 | 51.4 | 50.9 | 8101 | Maluku Tenggara Barat | 11.2 | 19.8 | 23.6 | 23.6 |
| 3308 | Magelang | 50.5 | 53.1 | 52.4 | 52.2 | 8102 | Maluku Tenggara | 37.2 | 46.2 | 33.2 | 32.1 |
| 3309 | Boyolali | 29.9 | 34.5 | 32.9 | 32.8 | 8103 | Maluku Tengah | 25.6 | 31.1 | 47.4 | 47.4 |
| 3310 | Klaten | 27.0 | 27.3 | 27.0 | 27.0 | 8104 | Buru | 19.4 | 21.7 | 24.1 | 24.5 |

**Table A1.** *Cont.*

| Code | District | 2014 | 2018 | 2019 | 2020 | Code | District | 2014 | 2018 | 2019 | 2020 |
|------|----------|------|------|------|------|------|----------|------|------|------|------|
| 3311 | Sukoharjo | 29.0 | 30.2 | 29.1 | 29.1 | 8105 | Kepulauan Aru | 0.0 | 1.1 | 2.6 | 2.5 |
| 3312 | Wonogiri | 45.2 | 46.7 | 46.5 | 45.8 | 8106 | Seram Bagian Barat | 18.0 | 29.1 | 42.5 | 41.7 |
| 3313 | Karanganyar | 31.6 | 31.9 | 33.4 | 33.2 | 8107 | Seram Bagian Timur | 0.7 | 3.1 | 7.7 | 7.5 |
| 3314 | Sragen | 32.5 | 34.5 | 34.0 | 34.2 | 8108 | Maluku Barat Daya | 0.0 | 5.2 | 3.6 | 3.5 |
| 3315 | Grobogan | 44.6 | 45.5 | 44.7 | 43.9 | 8109 | Buru Selatan | 2.9 | 7.0 | 9.3 | 9.3 |
| 3316 | Blora | 29.9 | 31.6 | 33.2 | 32.9 | 8171 | Ambon | 39.9 | 35.2 | 36.5 | 36.9 |
| 3317 | Rembang | 45.0 | 46.7 | 45.9 | 45.1 | 8172 | Tual | 24.5 | 35.9 | 35.6 | 35.1 |
| 3318 | Pati | 35.8 | 36.9 | 37.4 | 37.5 | 8201 | Halmahera Barat | 15.2 | 17.1 | 16.8 | 16.0 |
| 3319 | Kudus | 40.9 | 44.4 | 44.1 | 44.4 | 8202 | Halmahera Tengah | 29.0 | 48.8 | 45.7 | 44.8 |
| 3320 | Jepara | 16.7 | 19.7 | 20.6 | 20.4 | 8203 | Kepulauan Sula | 12.4 | 25.0 | 27.9 | 26.8 |
| 3321 | Demak | 35.5 | 36.9 | 36.7 | 36.6 | 8204 | Halmahera Selatan | 1.3 | 7.0 | 9.1 | 8.8 |
| 3322 | Semarang | 41.8 | 43.2 | 42.4 | 42.4 | 8205 | Halmahera Utara | 34.7 | 37.1 | 36.4 | 35.5 |
| 3323 | Temanggung | 46.1 | 47.5 | 46.4 | 46.1 | 8206 | Halmahera Timur | 14.4 | 38.5 | 38.1 | 37.1 |
| 3324 | Kendal | 35.0 | 36.1 | 35.7 | 36.1 | 8207 | Pulau Morotai | 19.8 | 41.4 | 49.6 | 48.7 |
| 3325 | Batang | 48.1 | 48.1 | 48.9 | 49.0 | 8208 | Pulau Taliabu | 0.0 | 0.0 | 10.5 | 10.4 |
| 3326 | Pekalongan | 42.7 | 42.5 | 41.5 | 41.3 | 8271 | Ternate | 67.4 | 69.4 | 35.5 | 34.3 |
| 3327 | Pemalang | 33.7 | 34.0 | 33.7 | 33.5 | 8272 | Tidore Kepulauan | 53.8 | 58.6 | 57.2 | 56.8 |
| 3328 | Tegal | 36.9 | 38.4 | 38.0 | 37.7 | 9101 | Fakfak | 11.1 | 22.7 | 13.7 | 13.9 |
| 3329 | Brebes | 44.0 | 43.4 | 46.4 | 46.1 | 9102 | Kaimana | 0.0 | 0.6 | 0.6 | 0.6 |
| 3374 | Semarang | 31.9 | 34.8 | 33.5 | 33.3 | 9103 | Teluk Wondama | 0.0 | 0.0 | 0.9 | 0.9 |
| 3375 | Pekalongan | 0.0 | 0.0 | 14.7 | 14.7 | 9104 | Teluk Bintuni | 1.4 | 1.7 | 1.7 | 1.9 |
| 3401 | Kulon Progo | 83.7 | 83.9 | 84.8 | 84.7 | 9105 | Manokwari | 54.6 | 54.8 | 55.8 | 56.1 |
| 3402 | Bantul | 84.4 | 85.0 | 84.6 | 84.7 | 9106 | Sorong Selatan | 0.7 | 2.4 | 0.8 | 0.7 |
| 3403 | Gunung Kidul | 66.5 | 66.9 | 65.9 | 65.8 | 9107 | Sorong | 18.6 | 29.9 | 34.5 | 34.0 |
| 3404 | Sleman | 67.0 | 68.5 | 67.8 | 68.1 | 9108 | Raja Ampat | 0.0 | 0.0 | 1.8 | 1.9 |
| 3501 | Pacitan | 31.1 | 55.8 | 50.6 | 49.4 | 9109 | Tambrauw | 0.0 | 0.0 | 0.2 | 0.2 |
| 3502 | Ponorogo | 31.4 | 33.0 | 32.5 | 31.5 | 9110 | Maybrat | 0.0 | 13.3 | 20.0 | 19.8 |
| 3503 | Trenggalek | 38.7 | 41.9 | 41.5 | 40.8 | 9111 | Manokwari Selatan | 0.0 | 0.0 | 40.2 | 39.0 |
| 3504 | Tulungagung | 9.7 | 19.3 | 19.3 | 19.1 | 9112 | Manokwari | 0.0 | 0.0 | 0.1 | 0.1 |
| 3505 | Blitar | 13.8 | 24.1 | 23.8 | 23.8 | 9171 | Sorong | 77.3 | 88.2 | 31.8 | 31.6 |
| 3506 | Kediri | 23.5 | 23.8 | 23.4 | 23.6 | 9401 | Merauke | 1.3 | 2.8 | 3.5 | 3.5 |
| 3507 | Malang | 23.2 | 23.7 | 23.3 | 23.0 | 9402 | Jayawijaya | 7.0 | 8.8 | 7.5 | 7.8 |
| 3508 | Lumajang | 22.0 | 32.4 | 32.9 | 32.5 | 9403 | Jayapura | 26.0 | 30.3 | 29.7 | 29.8 |
| 3509 | Jember | 26.4 | 27.2 | 27.2 | 27.1 | 9404 | Nabire | 19.1 | 24.6 | 23.4 | 22.1 |
| 3510 | Banyuwangi | 21.9 | 18.3 | 18.0 | 17.8 | 9408 | Kepulauan Yapen | 11.3 | 12.0 | 13.0 | 12.6 |
| 3511 | Bondowoso | 21.4 | 22.2 | 22.0 | 22.0 | 9409 | Biak Numfor | 32.3 | 40.4 | 40.6 | 39.7 |
| 3512 | Situbondo | 41.4 | 41.6 | 41.1 | 40.8 | 9410 | Paniai | 7.3 | 7.9 | 6.8 | 6.5 |
| 3513 | Probolinggo | 29.0 | 33.3 | 33.3 | 33.2 | 9411 | Puncak Jaya | 0.0 | 1.0 | 2.9 | 3.0 |
| 3514 | Pasuruan | 40.1 | 43.0 | 43.5 | 43.5 | 9412 | Mimika | 1.5 | 5.3 | 5.6 | 5.8 |
| 3515 | Sidoarjo | 19.2 | 24.5 | 24.9 | 24.9 | 9413 | Boven Digoel | 4.1 | 3.6 | 4.6 | 4.9 |
| 3516 | Mojokerto | 33.6 | 34.7 | 34.7 | 35.0 | 9414 | Mappi | 0.0 | 0.0 | 0.0 | 0.0 |
| 3517 | Jombang | 27.5 | 31.3 | 31.2 | 31.5 | 9415 | Asmat | 0.0 | 0.0 | 0.0 | 0.0 |
| 3518 | Nganjuk | 28.6 | 31.7 | 31.0 | 30.6 | 9416 | Yahukimo | 0.0 | 0.6 | 0.5 | 0.5 |
| 3519 | Madiun | 34.0 | 36.5 | 38.7 | 38.3 | 9417 | Pegunungan Bintang | 0.0 | 0.6 | 1.7 | 1.7 |
| 3520 | Magetan | 22.2 | 22.7 | 22.3 | 21.9 | 9418 | Tolikara | 0.0 | 0.5 | 2.4 | 2.4 |
| 3521 | Ngawi | 30.3 | 32.1 | 32.4 | 32.1 | 9419 | Sarmi | 3.4 | 14.1 | 15.8 | 16.0 |
| 3522 | Bojonegoro | 32.9 | 33.1 | 31.7 | 31.0 | 9420 | Keerom | 14.9 | 15.8 | 11.0 | 11.0 |
| 3523 | Tuban | 40.3 | 41.3 | 41.2 | 40.8 | 9426 | Waropen | 0.7 | 0.7 | 1.5 | 1.6 |
| 3524 | Lamongan | 30.4 | 30.7 | 31.0 | 31.3 | 9427 | Supiori | 40.1 | 40.3 | 40.4 | 39.7 |
| 3525 | Gresik | 26.3 | 26.1 | 26.2 | 26.1 | 9428 | Mamberamo Raya | 0.0 | 0.0 | 0.0 | 0.0 |
| 3526 | Bangkalan | 33.4 | 33.7 | 33.4 | 33.1 | 9429 | Nduga | 0.0 | 0.0 | 0.0 | 0.0 |
| 3527 | Sampang | 24.4 | 24.3 | 23.9 | 24.3 | 9430 | Lanny Jaya | 0.0 | 1.8 | 0.9 | 0.9 |
| 3528 | Pamekasan | 23.4 | 24.1 | 23.7 | 23.4 | 9431 | Mamberamo Tengah | 2.9 | 3.1 | 1.4 | 1.5 |
| 3529 | Sumenep | 24.1 | 24.4 | 23.8 | 23.5 | 9432 | Yalimo | 16.4 | 16.8 | 18.9 | 19.0 |
| 3574 | Probolinggo | 91.4 | 90.0 | 89.2 | 89.7 | 9433 | Puncak | 0.0 | 0.0 | 0.0 | 0.0 |
| 3579 | Batu | 45.0 | 47.1 | 45.8 | 44.8 | 9434 | Dogiyai | 4.3 | 5.6 | 8.2 | 7.9 |
| 3601 | Pandeglang | 30.5 | 32.6 | 33.0 | 32.6 | 9435 | Intan Jaya | 0.0 | 0.0 | 0.0 | 0.0 |
| 3602 | Lebak | 34.7 | 37.5 | 38.1 | 37.4 | 9436 | Deiyai | 0.1 | 3.9 | 8.3 | 8.3 |
| 3603 | Tangerang | 5.7 | 5.7 | 5.9 | 6.1 | 9471 | Jayapura | 59.0 | 62.0 | 62.1 | 62.8 |
| 3604 | Serang | 23.1 | 28.3 | 30.0 | 30.6 | | | | | | |

Source: Author's calculation.

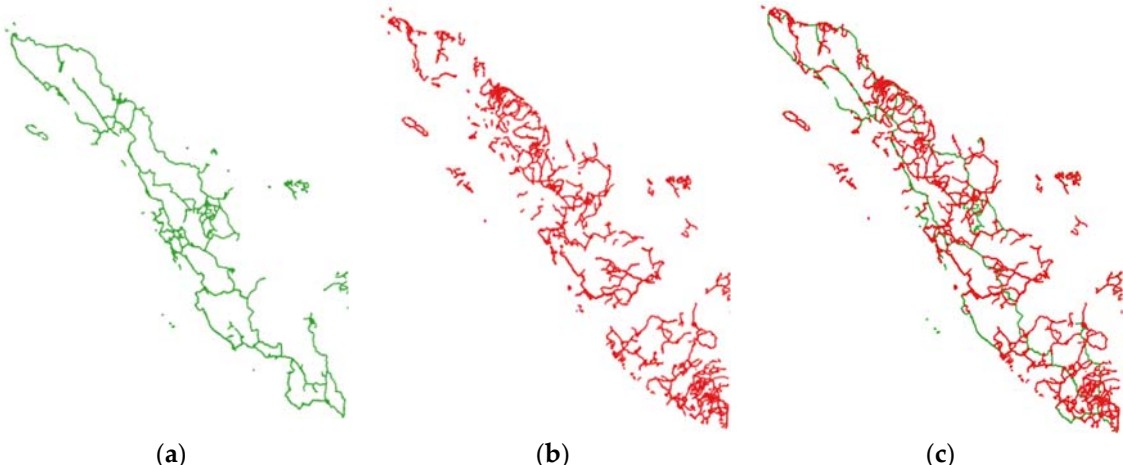

(**a**)                              (**b**)                              (**c**)

**Figure A1.** OSM data inconsistency (**a**) 2014 (**b**) 2020 (**c**) map merger. Note: Authors only use primary, primary link, secondary, and secondary link road classifications. Source: www.geofabrik.de (accessed on 15 November 2021).

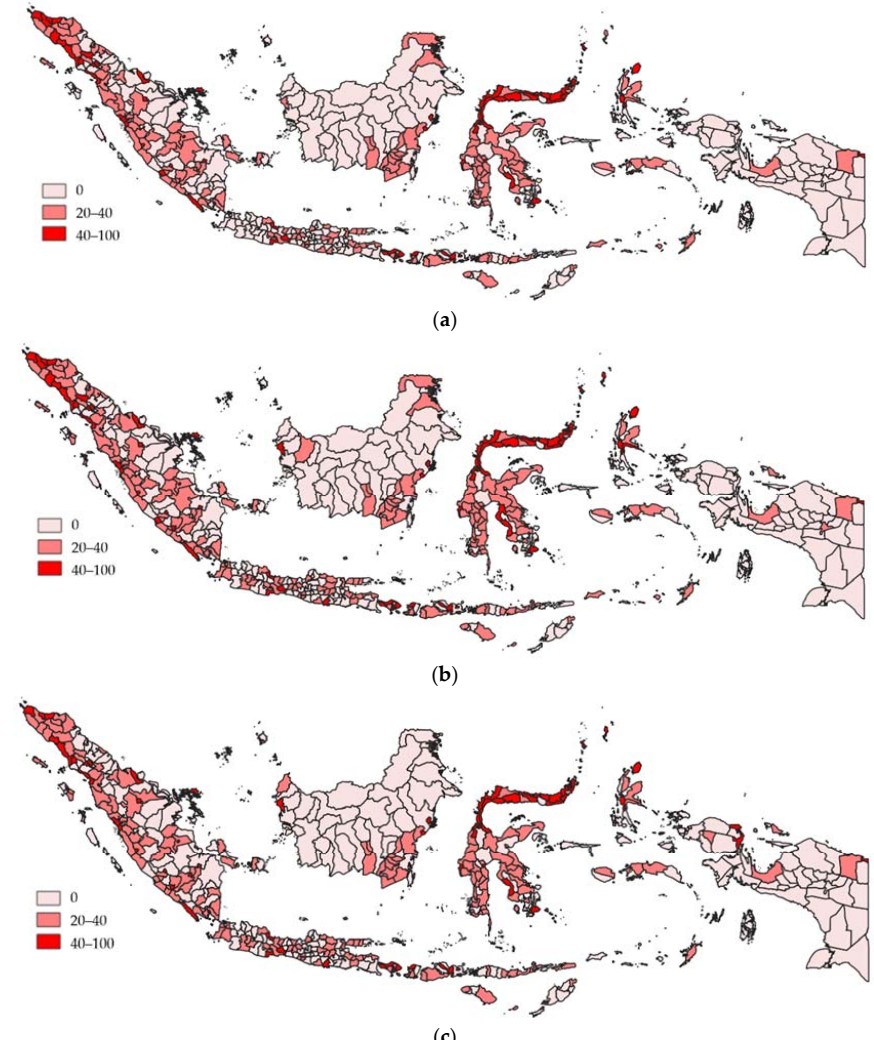

(a)

(b)

(c)

**Figure A2.** 2018 Indonesian RAI using the national road network map, WorldPop, and different road network condition data (per cent): (**a**) IRI (**b**) Road condition (**c**) Podes. Source: Author's calculation.

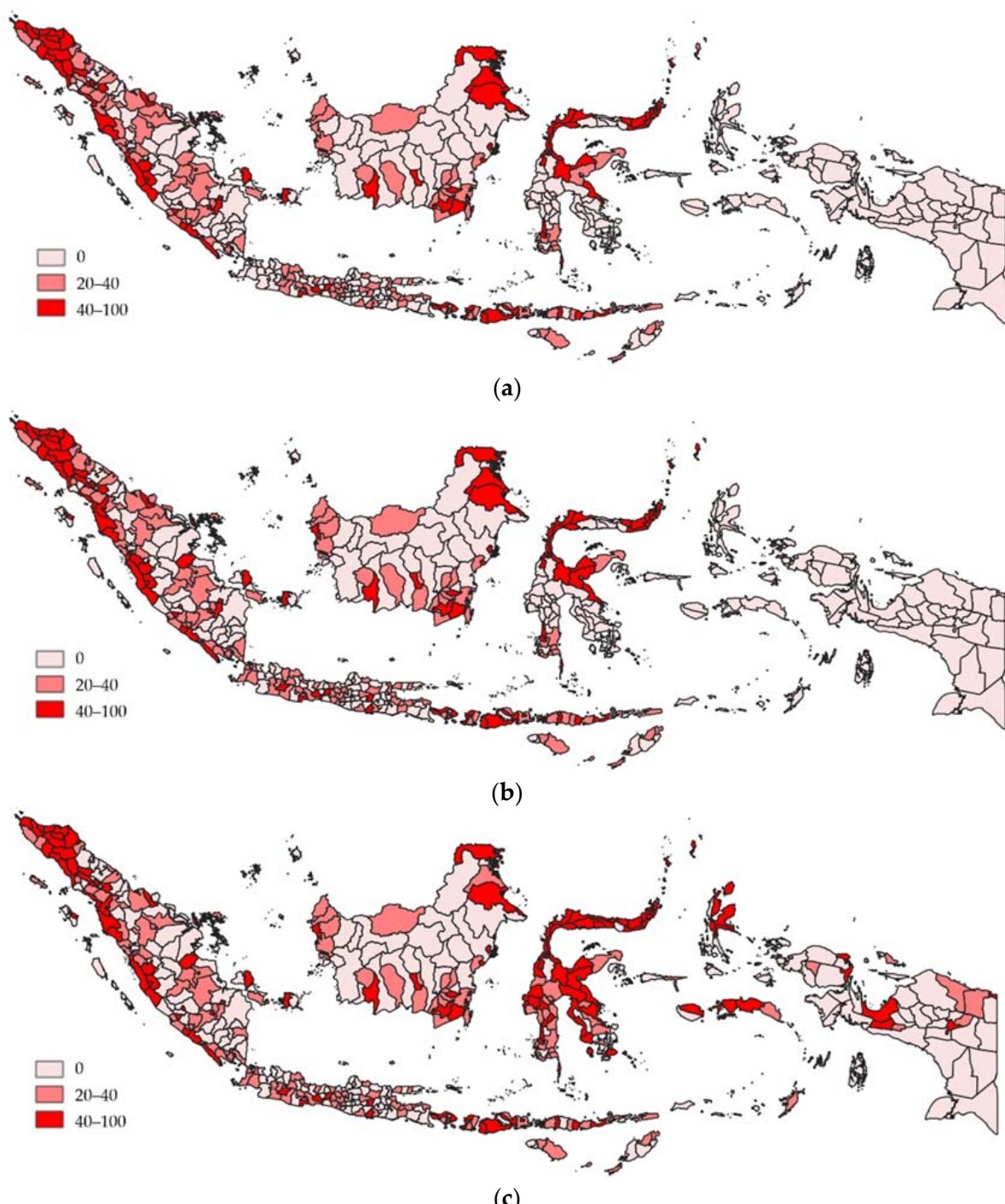

(**a**)

(**b**)

(**c**)

**Figure A3.** 2018 Indonesian RAI using the national road network map, LandScan, and different road network condition data (per cent): (**a**) IRI (**b**) Road condition (**c**) Podes. Source: Author's calculation.

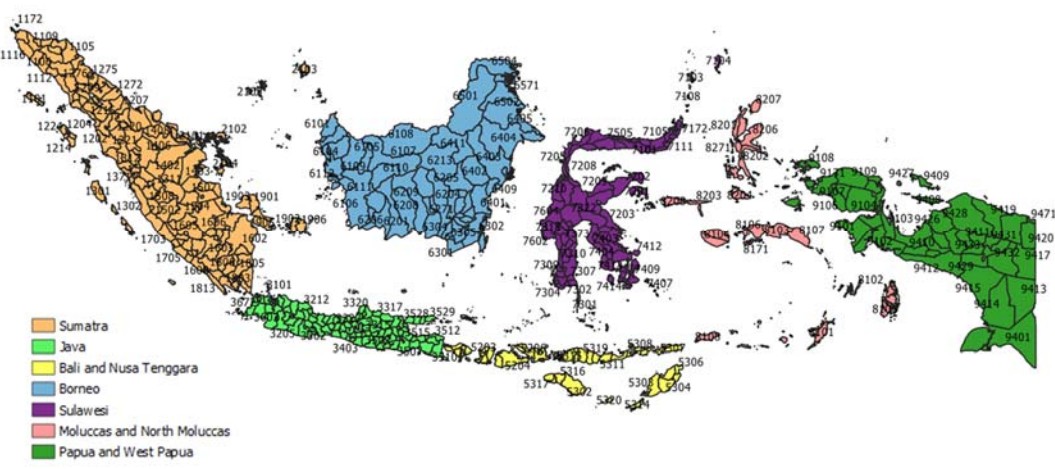

**Figure A4.** Indonesia by regional group and district's code.

## Notes

[1]    Bartlett's equal-variances test had $\chi^2(2)$ = 0.1065 and *p*-value = 0.948.

[2]    Sumatra has district codes 1101–2172, Java has district codes 3201–3673, Bali and Nusa Tenggara have district codes 5101–5371, Borneo has district codes 6101–6571, Sulawesi has district codes 7101–7606, Moluccas and North Moluccas have district codes 8101–8272, and Papua and West Papua have district codes 9101–9471.

[3]    According to Ministry of Finance Regulation Number 120/PMK.07/2020 about Regional Fiscal Capacity Maps, $DFCI_i = \frac{DFC_i}{\sum_{i=1}^{n} DFC_i/n}$ where $DFC_i$ is government revenue—(government revenue that its alocation is determined + specific expenditure) and *n* is the number of districts in Indonesia. $DFC_i$ shows the fiscal capacity of district *i*.

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
