# Peer review of "Inter-District Road Infrastructure and Spatial Inequality in Rural Indonesia"

_economies, doi:10.3390/economies10090229_

Round 1
Reviewer 1 Report
The paper aims at assessing the accessibility to all-season roads in rural Indonesia using Rural Access Index (RAI). The analysis is carried out for local districts, for which the appropriate date was not available. Hence, the first step of the study was to combine various data and maps to obtain desegregated information and corresponding maps for population distribution, urban-rural boundaries, and road network. Then the RAI was calculated and used to compere regional groups as well as measure the inequality of rural road accessibility across country.
The article is divided into sections: Introduction, Methodology, Results and Discussion, Conclusion and Recommendations, with added Appendix containing extra tables, maps and figures. The Introduction shortly states the motivation and the aim. It includes an elementary literature review. The most extensive part is the Methodology. It presents the step-by-step process of obtaining necessary information on rural locations, population, and roads. Moreover, available datasets and measures of inequality are introduced. The Results and Discussion section contains: the robustness check by using different data sources with the criteria for the final choice of dataset, analysis of road access inequality (by districts) aggregated to regional groups, as well as some demographic background and conclusions. Final section gives a summary of the study and its outcomes with some recommendation for the policy makers.
The study is very interesting and I think there is a demand for it, as road accessibility is a major problem in rural areas in developing regions in the world. It is also a physical barrier for economic growth and socioeconomic equality. The biggest value added of the study is the combining and comparing various data sets and maps to carry out the analysis on a micro level. This also allows for a sound robustness check of the outcomes. Unfortunately, the advantage of obtaining RAI on a micro-level is not highlighted in the paper. All results are presented in an aggregated form for regional groups. Maps presenting the RAI on the district level are available only in the Appendix Figure A1 & Figure 2 (it probably should be A2). It would be advisable to input at least one map (corresponding with the final selection of data sources) with proper inference on the spatial distribution of RAI in the Result section. Overall, the actual result of the study (RAI and inequality measures; subsection 3.2) are more scarily presented than the robustness check and data selection. The latter, described in detail in subsection 3.1 and substantial part of the Metrological section as well, appears to be the main focus of the paper.
The Results subsection 3.3 is titled Policy Implications, but it gives some discussion on demographical changes and conclusion of the study. It would be advisable to change the title of the subsection or transfer this to the Conclusion section.
The part of the study that needs to be improved the most is the Conclusion and Recommendations. The section is rather short and duplicates statements presented previously. The first paragraph (lines 399-409), constituting a third of the whole section, is a direct repetition of statements in the Introduction. It is unnecessary as it does not give any new information or perspective. Moreover, as declared in Conclusions (and the Introduction) the novelty of this study are the RAI results for districts, but, again, they are not directly present in the paper (only by regional groups), so it is difficult to appreciate the added value of this study. Additionally, lines 421-424 of the final paragraph in Conclusions duplicate statement form lines 274-378 in Results.
This study is design to indirectly asses the Indonesian policy on road infrastructure and this aim is achieved. In particular, Nawacita program is often referred to. It would be prudent to explain shortly what the Nawacita program is, preferably in the Introduction. Additionally, it would be beneficial to add in the Introduction (or in notes) brief information about Indonesian administrative division: districts, provinces and, particularly, regional groups (as they are important dimension of the results presentation).
There also some minor issues:
1. Figure 2 should be placed below the paragraph with first reference. (below line 137)
2. Line 170: non-decreasing order means ??? is larger or equal ???−1 (“≥” in brackets).
3. Lines 217-222: explain shortly β convergence (as σ convergence is explained)
4. Line 238: “A higher coefficient ? corresponds to a greater tendency for convergence.” while −1 < ? < 0 (line 236) indicates that the closer ? is to 0 (its maximal value), the grater the convergence. I think this is not the intended conclusion. Maybe “A higher absolute value of coefficient ? …”. Secondly, ? can be positive, indicating divergence. It would be prudent to mention it (in the Methodological section), as it is hypothetically possible.
5. Line 296: “WorldPop” instead of “Worldpop”.
6. I don’t see in the text any reference to Figure 6 and Table 5. Results of Table 5 are presented (lines 308-313), but Figure 6 regional values of median RAI are not interpreted in the text, just mentioned (line 337). It would be good to add 1-2 sentences interpreting the levels of RAI across region groups (for instance in the paragraph lines 308-313 or 336-339)
7. Paragraph in lines 336-339 needs reference, probably to Figure 6 and Table 5 (see previous comment).
8. Line 348: I don’t think 20% is “negligible”. Maybe you could phrase it a bit more cautiously.
9. Line 369: “On the contrary” in the middle of the paragraph seems to negate all previous statements about years 2014-2018, instead of indicating a change in pattern over next years. Maybe “on the other hand…” or “in the following years …” would better convey the reversed trends for rural populations and rural populations living near an all-season road.
10. There is no reference to Table 7 showing the simulation of all rural road being all-seasonal. It is briefly and indirectly interpreted in lines 375-379, but a reference and a more direct explanation would be appreciated.
11. Line 418: “Besides, the contribution of within-regional inequality has decreased consistently” this is a bit of an over statement. The decrease between 2014 and 2020 was of around 2 percentage points (Fig.8), that’s a marginal decrease over 7-year period. Especially, that you considered almost 20% of between-region inequality “negligible” (line 348).
Author Response
We would like to thank the reviewers for their comments, which we feel have helped us improve our article. All major changes, as responses to reviewers' comments, have been marked with our comments in the manuscript text (please see the attachment in pages 25-27). Lastly, we hope our revisions are in line with the reviewers' requests.

Reviewer 2 Report
Referee Report on “Inter-District Road Infrastructure and Spatial Inequality in Rural Indonesia”
Summary
This paper examines the accessibility of rural population in Indonesia to “all-season” roads and evaluate the inequality of such accessibility within and across regions in the country. The authors computed the Rural Access Index (RAI), a measure of rural population within a two-kilometer buffer of all-season roads, using various data sources about road network conditions and population distributions. The authors evaluated the inequality of rural road accessibility with three measures, the coefficient of variance, the Gini index, and the Theil index, and examined β and σ-convergence of RAI. The authors drew conclusions that rural road accessibility increased, inequality decreased, and a significant trend of convergence during the period of 2011-2020 is observed. My overall evaluation of this paper is that the topic is interesting, and the methods are appropriate, but the authors failed to clearly convey their thoughts and wrote the paper in a disordered and hasty manner. Therefore, my recommendation is to substantially revise the paper in an academic and rigorous way; otherwise, a rejection can also be rendered.
Following are my main critiques to this paper.
1. My reading experience left me an impression that the authors did not have the patience to explain in a reasonable length some key concepts, equations, and results shown in tables and figures. Here are a few such incidences:
o The construction of the RAI is an allegedly main contribution of this work. However, the authors just wrote about the computation of the RAI with one sentence: “Step-by-step procedures for calculating the RAI are shown in Figure 1.” (line 88-89, page 3) Although Figure 1 looks good, some narratives are still needed at least to explain the process shown in the figure. The two or three paragraphs that follow only explain the data on population distribution, urban-rural delimitation, and road network conditions, not on the computation of the RAI per se.
o The authors used both β and σ-convergence to assess whether road access inequality has attenuated since the implementation of the Nawacita program. The authors estimated equation (8) to examine β-convergence, but what about σ-convergence? σ-convergence is not just equation (7) but a difference equation to be estimated. The authors did not explain how they evaluated σ-convergence. As for the estimated result of β-convergence, it is only shown in footnote 2 in page 12. If evaluating β-convergence is a truely important part of this paper, the authors should display the results in a more apparent way, such as a table, and show standard statistics, such as standard errors, p-values, goodness-of-fit, etc. Finally, where are the results of σ-convergence shown, any figure or table for that? I can only find two sentences for that in page 11-12, which is far from enough for a clear explanation. By the way, what is the Nawacita program? The authors should explain it briefly in the introduction section.
o Tables and figures are supposed to be the most eye-catching elements in a paper. They should be well explained and clearly referred in the main text. Table 7 shows the “real condition and simulation of inequality indicators in 2014–2020”. How are the numbers in the table generated? I cannot find any explanation and reference to this table anywhere in the manuscript. Where are the references to tables 1 and 5 and figures 4 and 9?
2. The authors chose some unimportant tables and figures in the main text but left really important ones in the appendix. This may be a matter of personal taste. But as far as my writing experience concerned, I would put tables and figures of my main contribution in the main text and put those for a mere reference purpose and robustness checks in appendix. In this paper, Table 2 and figures 2 and 5 are merely references to official or the OpenStreetMap data that are not authors’ own contribution; on the other hand, Table A1 and Figure A1 are their own work and show the main results for the computation of the RAI, which should be put in the main text. Figure 3 is also not that important, which only shows the standard way to compute the Gini index from a Lorenz curve. Relatedly, unimportant equations should not be put in a displayed or numbered mode, like equation (1) and (5). In equation (6), what is wkt? The authors explained in length what the Gini index was, but they did not give an equal treatment to the Theil index. Before explaining the decomposition of the Theil, a brief explanation of what the Theil is and how it measures inequality is necessary. Overall, the authors need to rethink what are the really important elements of this paper to be conveyed to readers.
3. Finally, although it is a cliché to recommend non-native English speakers to pay attention to the writing style, I do have to suggest the authors choose appropriate words, correct grammatic errors, use more active instead of passive voice, and possibly find a native English speaker for a proof-reading.
Author Response
We would like to thank the reviewers for their comments, which we feel have helped us improve our article. All major changes, as responses to reviewers' comments, have been marked with our comments in the manuscript text (please see the attachment in pages 25-26). Lastly, we hope our revisions are in line with the reviewers' requests.

Reviewer 3 Report
Dear author(s)
I highly appreciate this study regarding the issue of road infrastructure and spatial inequality in Rural Indonesia.
However, I have simple comments:
1. The abstract should provide more information regarding the findings, policy implementation, and the scientific contribution of the study.
2. The discussion of the effect of road infrastructure on spatial inequality may be completed with the information or the data on fiscal capacity in Indonesia.
3 The conclusion may highlight the specific policy, especially for the local government to improve the quantity of road infrastructure to reduce spatial inequality.
Author Response
We would like to thank the reviewers for their comments, which we feel have helped us improve our article. All 754 major changes, as responses to reviewers' comments, have been marked with our comments in the manuscript 755 text (please see the attachment in pages 25). Lastly, we hope our revisions are in line with the reviewers' requests.

Round 2
Reviewer 2 Report
The authors have significantly improved the quality of papers. I recommend acceptance of the paper. Some minor revisions are needed.
1. On page 5, should Yit is ordered by time or by region? To plot the Lorenz curve and compute the Gini index, regions should be first sorted from the least to the greatest. So, I think it should be Yit >= Y(i-1)t.
2. In equation 6, the coefficient on ln(RAIit) should just be beta not the fraction that includes beta in a non-linear way unless the author did mean to estimate beta in a non-linear model. Please clarify this.
3. The paragraph under the header 3.2 on page 8 is written redundantly. The authors could put the district numbers in a footnote.
Author Response
We would like to thank you for your comments and suggestions, Sir. These have helped us improve our article. All minor changes have been marked with our comments in the manuscript text (please see our explanations in page 25). Lastly, we hope our revisions are in line with your requests.
